# Genomic language model mitigates chimera artifacts in nanopore direct RNA sequencing

Yangyang Li [1,3], Ting-You Wang [1,3], Qingxiang Guo [1], Yanan Ren[1],
Xiaotong Lu[1], Qi Cao [1,2] & Rendong Yang [1,2] ✉

Chimera artifacts in nanopore direct RNA sequencing (dRNA-seq) introduce substantial inaccuracies, complicating downstream applications such as transcript annotation and gene fusion detection. Current basecalling models are unable to detect or mitigate these artifacts, limiting the reliability and utility of dRNA-seq for transcriptomics research. To address this challenge, we present DeepChopper, a genomic language model specifically designed to identify and remove adapter sequences from base-called dRNA-seq long reads with single-base precision. Operating independently of raw signal or alignment information, DeepChopper effectively eliminates adapter-bridged artifacts. Here, we show that DeepChopper enhances the accuracy of downstream analyses and unlocks the full potential of nanopore dRNA-seq, establishing it as a more robust tool for diverse transcriptomics applications.

Long-read RNA sequencing technologies are revolutionizing transcriptomic research by providing unparalleled resolution for detecting complex splicing and gene fusion events often missed by conventional short-read RNA-seq methods. Among these technologies, Oxford Nanopore Technologies (ONT) dRNA-seq stands out by sequencing full-length RNA molecules directly, preserving native RNA modifications and allowing a more accurate and comprehensive analysis of RNA biology. This approach bypasses the inherent limitations of cDNA-based sequencing methods, such as artifacts arising from reverse transcription (RT), template switching, and polymerase chain reaction (PCR) amplification[1–3].

Despite these advantages, a critical question remains: Does ONT dRNA-seq introduce technical artifacts? A previous study has suggested that dRNA-seq might generate chimera artifacts, leading to multi-mapped reads[4], but systematic characterization of these artifacts (their prevalence, formation mechanisms, and persistence with modern sequencing chemistries such as RNA004 and current Dorado basecalling[5]) remains limited in peer-reviewed literature. These artifacts may result from ligation during library preparation or chimeric reads produced by software missing the open pore signal, potentially confounding downstream analyses such as transcriptome assembly, quantification, and detection of alternative splicing and

gene fusion events. Detecting these chimera artifacts is challenging because long-read aligners often produce chimeric alignments from such artifacts that are indistinguishable from those derived from true gene fusion events. Importantly, chimera artifacts frequently contain internal adapter sequences[4], suggesting that these adapter-bridged chimeras could theoretically be distinguished from biological chimeras by detecting the presence of internal adapters. However, ONT dRNA-seq basecallers, trained in RNA, struggle to properly call these DNA-based adapter sequences under an RNA model[6]. As a result, current adapter detection tools[7–9] cannot exploit this feature to eliminate these adapter-bridged chimeras, leaving the issue unresolved (Supplementary Table 1).

To address this unmet need, we developed DeepChopper, a genomic language model (GLM) for long-read sequence analysis. Leveraging recent advances in large language model (LLM) that can interpret complex genetic patterns[10], DeepChopper processes long genomic contexts with single-nucleotide resolution. This capability enables precise identification of ONT adapter sequences within base-called long reads, facilitating the detection and removal of chimera artifacts in dRNA-seq data. Through analysis of both existing and newly generated dRNA-seq data, including those using the most recent RNA004 chemistry, we uncovered the prevalence of chimera artifacts,

[1]Department of Urology, Northwestern University Feinberg School of Medicine, Chicago, IL, USA. [2]Robert H. Lurie Comprehensive Cancer Center, Northwestern University Feinberg School of Medicine, Chicago, IL, USA. [3]These authors contributed equally: Yangyang Li, Ting-You Wang.
✉e-mail: rendong.yang@northwestern.edu

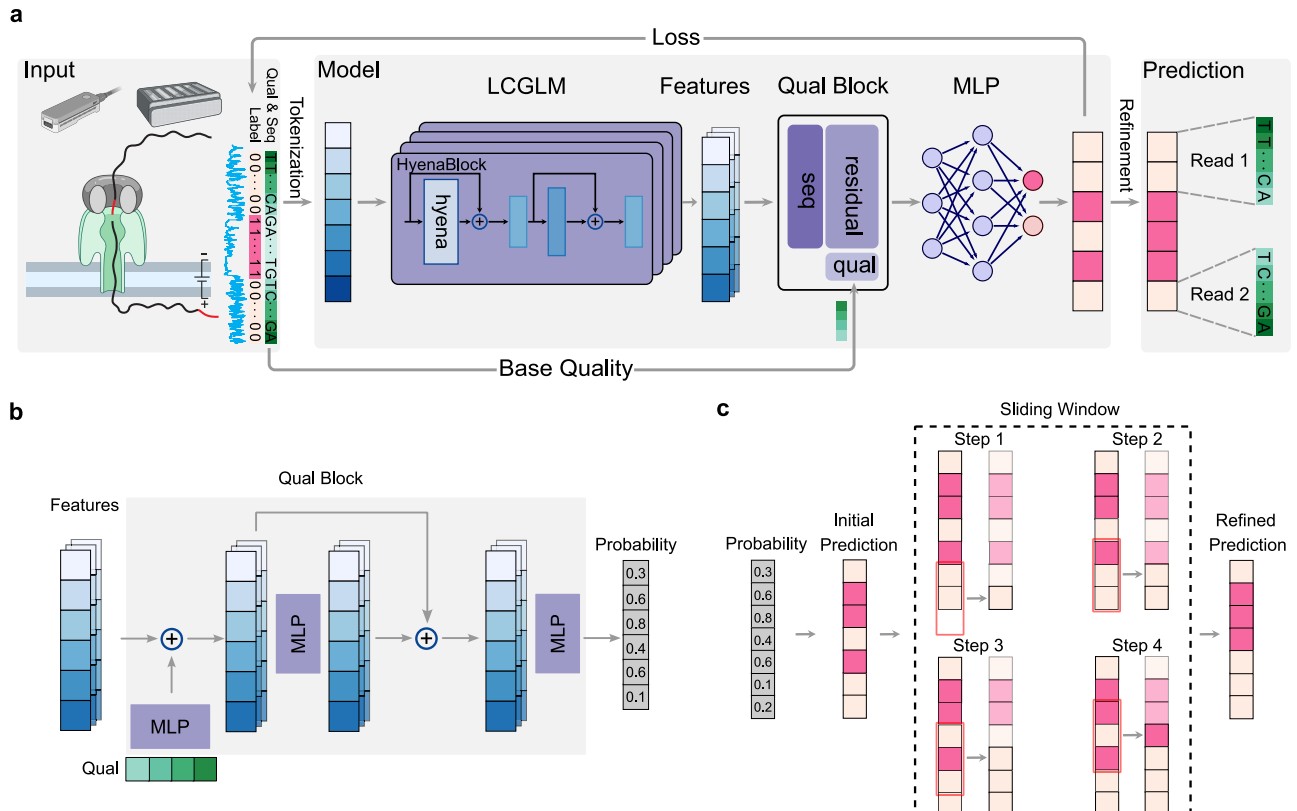

**Fig. 1 | DeepChopper architecture and methodology. a** Overview of the Deep-Chopper model. Raw sequences are first tokenized into vectors and processed by HyenaDNA, a long-context genomic language model (LCGLM), to generate embedding features. These features are integrated with base quality information in the quality block to produce per-token probability scores. A refinement strategy further optimizes the predictions. Created in BioRender. Guo, Q. (2026) https://BioRender.com/zlls0nk. **b** Architecture of the quality block. The block combines a multi-layer perceptron (MLP) (purple) with a residual connection to process both embedding features and sequence base quality scores (green vector). The output provides per-token probabilities indicating whether each base belongs to an adapter sequence. **c** Illustration of the sliding window refinement method. The model's initial predictions are inferred from probability. Then the predictions are processed using a sliding window approach (red rectangle) to refine predictions. The dashed rectangle highlights the first four steps of this refinement process, where each step refines the prediction for a single base position in terms of the majority vote.

a critical issue previously overlooked in the long-read sequencing field. We demonstrated that these artifacts significantly impact transcriptomic analysis by complicating gene fusion detection, transcript annotation, and alternative splicing studies. By both identifying and addressing this problem, our work enhances the reliability and precision of dRNA-seq data, improving its utility in transcriptomic research.

## Results

### DeepChopper architecture and training

DeepChopper leverages the long-context genomic language model (LCGLM) HyenaDNA[11], which excels at capturing long-range dependencies (Fig. 1a). To process sequencing base quality information, DeepChopper extends its framework by incorporating a dedicated quality block, which is a neural network comprising multiple multi-layer perceptrons (MLPs) with residual connections[12] (Fig. 1b, See Methods for details). This addition enables the effective utilization of sequencing base quality, a crucial feature for improving prediction accuracy, particularly for distinguishing genuine adapter sequences from similar motifs that may occur naturally within reads or from low-quality regions. By combining broad contextual understanding with nucleotide-level precision, this hybrid architecture allows DeepChopper to accurately identify and process adapters sequences. Reads containing internal adapters are split into multiple records, with 3′ end adapters simultaneously trimmed (Fig. 1a), thereby preserving authentic biological sequences and eliminating chimera artifacts.

To further refine the prediction accuracy, DeepChopper implements a post-processing stage using a sliding window and majority vote approach, as illustrated in Fig. 1c. The model applies a sliding window with a stride of 1 across the read, analyzing the distribution of predicted adapter positions within each window (See Methods for details). This refinement process operates on the initial predictions independently for each position, ensuring that each base's final classification reflects an aggregation of local context without error propagation. By maintaining precise boundary detection at single-nucleotide resolution, this strategy ensures that predicted adapter sequences correspond to biologically plausible boundaries, enabling accurate splitting of chimeric reads into their component sub-reads while minimizing spurious fragmentation.

Comparing to existing general-purpose GLM, DeepChopper is specifically optimized for long-read sequence analysis at single-nucleotide resolution. This fine-grained resolution provides a critical advantage for genomic analysis tasks requiring precise base-level predictions. While DNABERT[13] is limited to input sequences of approximately 512 bp, DNABERT2[14] to 10,000 bp, and Nucleotide Transformer[15] to 6000 bp, DeepChopper supports input lengths up to 32 kilobases, sufficient to encompass most complete mRNA transcripts. This 32K nucleotide input limit was selected based on both technical and biological considerations. Technically, the constraint reflects the architectural design and context window limitations of the underlying HyenaDNA model. Biologically, human protein-coding transcripts have a median length of approximately 2.7 kb (95th

percentile: ~8.7 kb, 99th percentile: ~14 kb), with over 99.97% of transcripts falling below 32 kb based on current annotations (Supplementary Fig. 1). Empirical analysis of our dRNA-seq datasets confirmed that only 0 −0.0032% of reads exceeded this threshold across multiple cell lines and chemistries, with mean read lengths ranging from 683 to 1148 bp (Supplementary Table 2). This extended input capacity, combined with single-nucleotide tokenization, enables DeepChopper to accurately identify non-reference elements such as ONT adapter sequences with base-pair precision, an essential capability for detecting and recovering adapter-bridged chimeras in dRNA-seq data.

In addition, DeepChopper's lightweight architecture, consisting of only 4.6 million parameters, makes it computationally efficient and scalable for large-scale dRNA-seq analysis. This is in contrast to models like Evo[16], which require billions of parameters and more computational resources.

To train DeepChopper for identifying adapter sequences within dRNA-seq long reads, we utilized data from six human cell lines: HEK293T, A549, HCT116, HepG2, K562 and MCF-7 provided by the Singapore Nanopore Expression Project (SG-NEx)[17] (Supplementary Table 2). We curated a training set of 480,000 long reads and a validation set of 60,000 ones initially deemed free of adapters and inserted putative adapter sequences, derived from the raw dRNA-seq data, into these reads to create instances containing internal and 3′ end adapters (See Methods for details). An independent test set comprising 60,000 long reads was held out for performance evaluation.

## DeepChopper benchmarking and model optimization

We conducted comprehensive benchmarking of DeepChopper against existing ONT adapter trimming tools including Pychopper[7], Porechop[8], and Porechop_ABI[9], though it should be noted that none of these existing tools were specifically designed for dRNA-seq data analysis. Performance evaluation was carried out using the synthetic testing dataset ($n = 60,000$ reads), enabling rigorous assessment of precision, recall, and F1-score metrics. As shown in Supplementary Fig. 2, all existing tools demonstrated negligible performance metrics when processing dRNA-seq adapter sequences, indicating fundamental incompatibility with the dRNA-seq protocol. In contrast, DeepChopper achieved exceptional accuracy in identifying both terminal and internal adapters, with recall, precision, and F1 scores consistently exceeding 0.99. These results highlight DeepChopper's unique capability to address the specific complexities inherent to dRNA-seq reads and underscore the critical need for purpose-built solutions in this domain.

In addition to adapter-trimming tools, we also evaluated Breakinator[18], which is designed to detect chimeric artifacts rather than adapter boundaries. Breakinator was benchmarked separately using real VCaP RNA002 dRNA-seq data. As shown in Supplementary Fig. 3, Breakinator reduced detected artifacts by 62% but simultaneously decreased cDNA support to 4.8% (baseline 5.8%), reflecting substantial loss of biologically supported signal. In contrast, DeepChopper reduced artifacts by 91% while increasing cDNA support to 49%, demonstrating selective removal of true artifacts while preserving biological transcript complexity. A substantial fraction of dRNA-seq artifacts do not appear as chimeric alignments or fail to map (Supplementary Fig. 4), making them inherently undetectable by alignment-rule-based chimera callers.

To further evaluate DeepChopper's classification accuracy, we assessed its performance at single-nucleotide resolution. As shown in Supplementary Fig. 5, the model exhibited sharply bimodal probability distributions for base-level classification, stratified by ground truth labels. For true adapter bases (Supplementary Fig. 5a), the model consistently assigned high probabilities ( >0.8) to the adapter class while suppressing probabilities for the non-adapter class ( <0.2). Conversely, for true non-adapter bases (Supplementary Fig. 5b), the model reliably predicted high probabilities for the non-adapter class and low probabilities for the adapter class. The minimal presence of intermediate probability values suggests that DeepChopper makes highly confident predictions with low ambiguity between classes. This decisive classification behavior reflects the model's robust feature representation, well-calibrated decision boundaries, and strong generalization to previously unseen data.

We further conducted an ablation experiment to assess the contribution of the quality block component in the model architecture. As shown in Supplementary Table 3, inclusion of the quality block led to a marked improvement in performance, with the F1 score increasing from 0.97 to 0.99. This module leverages per-base quality scores to better distinguish genuine adapter sequences from similar motifs that may occur naturally within reads. The enhanced performance suggests that incorporating sequencing quality information enables the model to more effectively filter out spurious signal and improve classification robustness. This experiment was performed using the same training methodology as previous evaluations but with a newly generated, focused dataset of 100,000 reads (See Methods for details).

## Chimeric read artifact detection in cancer dRNA-seq data

To assess DeepChopper's ability to detect chimera artifacts in real data, we generated an independent dRNA-seq dataset using the prostate cancer VCaP cell line, which was excluded from model training. This dataset provides a robust framework for evaluating chimera artifacts in genuine dRNA-seq samples, ensuring that DeepChopper's performance generalizes beyond the training data. We conducted dRNA-seq of VCaP cells using ONT's SQK-RNA002 chemistry, consistent with that used in the SG-NEx project. Using a MinION sequencer with four R9.4 flow cells, we generated 9,177,639 long reads in FASTQ format, with base-calling performed using SG-NEx's Dorado software[5].

DeepChopper processes input data through a three-stage pipeline: (1) FASTQ-to-Parquet conversion for efficient input/output, (2) adapter prediction using a neural network, and (3) post-processing to trim and segment reads based on predicted adapter positions. To improve runtime, we implemented core functions in Rust, enabled GPU-based inference, and parallelized key components across the pipeline.

We systematically benchmarked DeepChopper's computational performance to assess scalability for large-scale dRNA-seq studies, testing datasets ranging from 0.1M to 23M reads. Using VCaP read subsamples from 0.1M to 9M reads, we observed near-linear runtime scaling (Supplementary Fig. 6a), with the 9M VCaP dataset requiring approximately 5 h on two NVIDIA A100 GPUs. Memory usage increased with input size, peaking at approximately 93 GB for CPU-based inference and 56 GB for GPU-based execution (Supplementary Fig. 6b). To evaluate performance at substantially larger scale, we benchmarked a merged dataset of 23 million reads combining five cell lines (A549, HCT116, HepG2, K562, and MCF7), which required approximately 10.6 h total processing time (23 min for FASTQ conversion, 8.5 h for prediction, 1.7 h for post-processing). Importantly, runtime continued to scale near-linearly and memory usage remained stable across the entire range, demonstrating that DeepChopper can process datasets larger than 23M reads with no fundamental computational barriers to scaling (See Methods for details).

Applying DeepChopper to the full VCaP dataset increased usable read yield by 3%, resulting in 9,357,913 adapter-trimmed reads. It identified 8,218,172 adapter sequences across 7,990,102 reads (87% of total), most measuring ~70 bp, consistent with the expected length of the RMX adapter used in ONT's SQK-RNA002 dRNA-seq kit (Fig. 2a)[19]. Analysis of adapter locations revealed that 7,777,624 reads had adapters at the 3′ end, while 148,452 contained internal adapters (Fig. 2b), indicating that chimeric artifacts are common in VCaP dRNA-seq data.

Further examination showed that chimera artifacts could arise from the joining of multiple long reads, with the most frequent pattern involving two reads joined by a single internal adapter (Fig. 2c). To

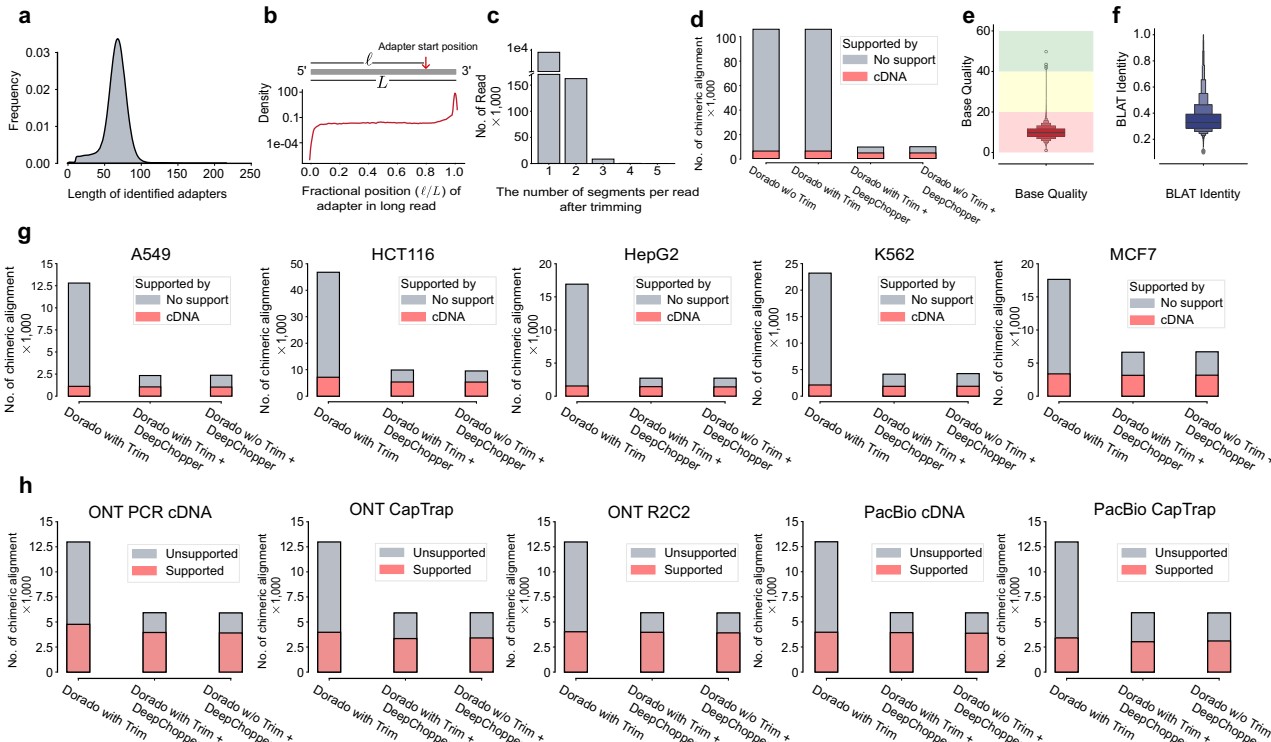

**Fig. 2 | Detection of chimeric read artifacts in dRNA-seq data using Deep-Chopper and validation with orthogonal sequencing platforms. a** Length distribution of predicted adapters by DeepChopper in VCaP dRNA-seq data. **b** Distribution of relative adapter position along read length in VCaP dRNA-seq data. Grey rectangle represents a long read from 5′ to 3′. Relative position is calculated as the ratio of the length before DeepChopper predicted adapter start position to the total read length. **c** Distribution of segments per read after trimming: 1 segment indicates 3′ end adapter trimmed, while 2 or more indicate internal adapters trimmed. **d** Chimeric alignments (in thousands) for VCaP dRNA-seq reads processed by Dorado with and without adapter trimming, Dorado with adapter trimming followed by DeepChopper, and DeepChopper. Gray bars represent unsupported chimeric alignments (likely artifacts); pink bars represent cDNA-supported chimeric alignments (biological events). DeepChopper-involved methods reduce chimeric alignments not supported by direct cDNA sequencing. **e** Base quality score distribution for internal adapter sequences identified by DeepChopper. Background colors indicate quality thresholds: green (Q > 20), yellow (Q = 10–20), red (Q <10). Enhanced box plot shows median (center line, 9.80), interquartile

range (innermost box, 7.97–11.74), and progressively more extreme percentiles (outer boxes). Adapter sequences exhibit low quality (mean Q-score = 9.96). $n = 146, 580$ internal adapter sequences. **f** BLAT identity distribution for internal adapter sequences aligned to the human reference genome (GRCh38). Box plot elements as in (**e**). Adapter sequences show poor identity (median = 0.33, IQR = 0.28–0.39, mean = 0.36). $n = 98,807$ sequences. **g** The number of chimeric alignments (in thousands) for A549, HCT116, HepG2, K562, and MCF7 cell lines processed by Dorado with adapter trimming, Dorado with adapter trimming followed by DeepChopper, and DeepChopper. Gray bars represent unsupported chimeric alignments (likely artifacts); pink bars represent cDNA-supported chimeric alignments (biological events). DeepChopper-involved methods consistently reduce chimeric alignments not supported by cDNA sequencing across all cell lines.
**h** Chimeric alignments from dRNA-seq of the WTC11 cell line, evaluated for support using additional ONT and PacBio sequencing data with different protocols. DeepChopper-involved methods reduce unsupported chimeric alignments across all methods compared to Dorado with adapter trimming.

validate these findings, we analyzed minimap2[20] chimeric alignments and compared them to a matched VCaP direct cDNA-seq dataset, which we generated as part of this study. Chimeric reads fully supported by cDNA sequencing were considered bona fide events (See Methods for details). Notably, we also evaluated whether ONT's Dorado basecaller trimming feature could mitigate these artifacts. However, we found that Dorado alone, regardless of trimming, was insufficient to eliminate spurious chimeric alignments. In contrast, DeepChopper reduced unsupported chimeric alignments by around 95% and increased the fraction of cDNA-supported chimeric events from 5.8% to 48.7%, whether applied before or after Dorado trimming (Fig. 2d). These results underscore DeepChopper's ability to distinguish true biological chimeras from technical artifacts.

To further verify the artifactual nature of internal adapters, we analyzed their base quality scores and aligned them to the human reference genome using BLAST-like alignment tool (BLAT)[21]. Adapter regions identified within chimera artifacts exhibited lower base quality (mean Q-score: 9.96) (Fig. 2e) and poor sequence identity (mean BLAT identity: 0.36) to the reference genome (Fig. 2f), supporting their non-human and non-biological origin.

Finally, we evaluated post-processing performance as a function of the sliding window size used for segmentation. Using a 1M-read subsample, we tested window sizes of 11, 21, 31, 41, and 51 bp. Smaller windows yielded slightly higher cDNA support percentages (47.5% for 11 bp vs. 43.3% for 51 bp; Supplementary Fig. 7a), but increased fragmentation of reads into 4+ segments (Supplementary Fig. 7b). A 21 bp window provided the optimal balance, maintaining high support while minimizing over-segmentation. Based on these results, 21 bp was selected as the default setting, and DeepChopper allows users to adjust this parameter for dataset-specific optimization (See Methods for details).

## Multi-sample validation across platforms and species
To further evaluate DeepChopper's performance beyond the VCaP dataset, we performed multi-sample validation across diverse biological systems and sequencing platforms. We began by analyzing dRNA-seq data from the SG-NEx project, comparing chimeric alignments before and after DeepChopper trimming. DeepChopper detected internal adapters in 0.67–1.25% of reads across these datasets (A549: 0.92%, MCF7: 0.67%, HCT116: 1.22%, K562: 0.96%,

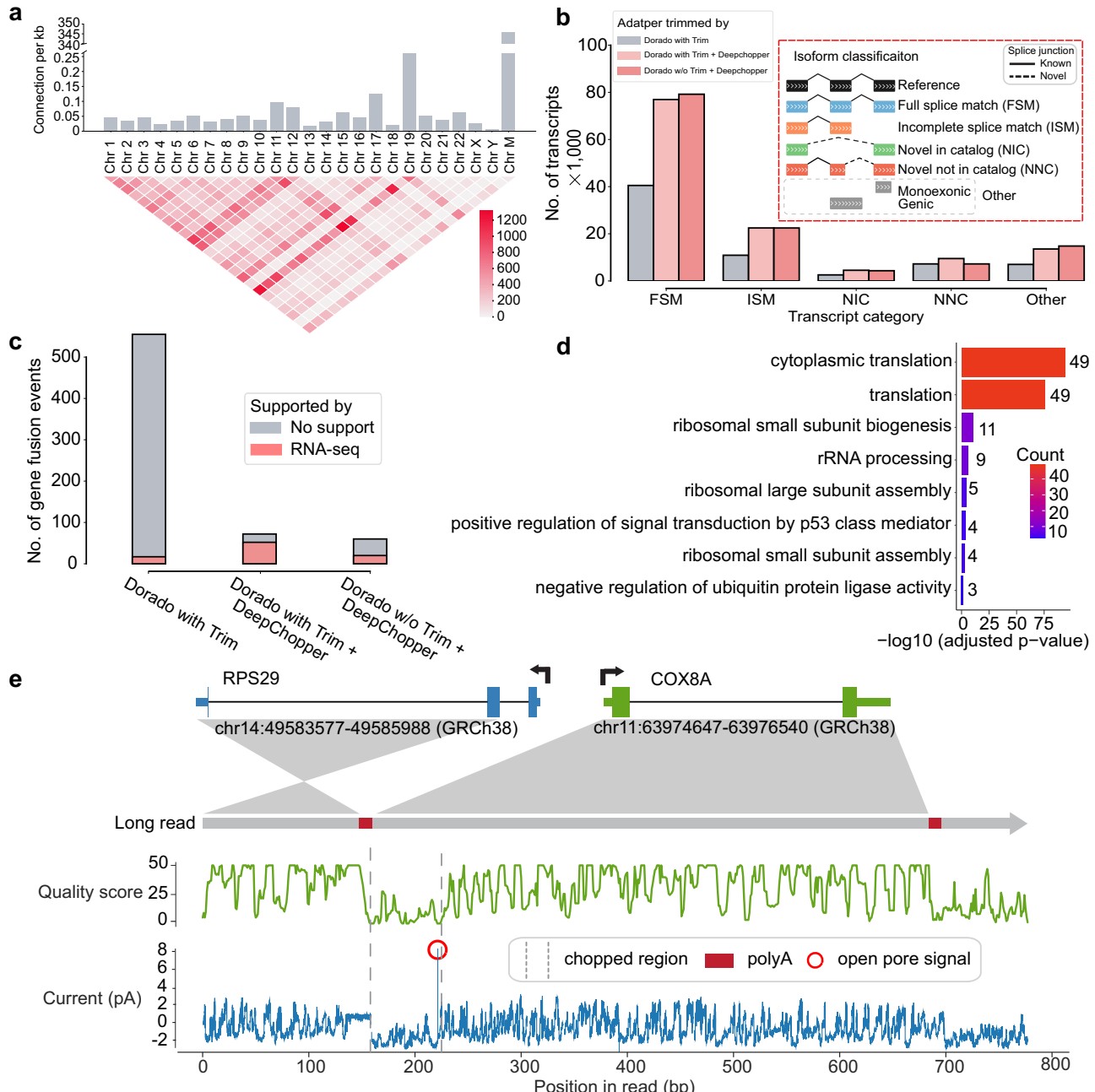

**Fig. 3 | Characterization of dRNA-seq chimera artifacts and their impact on downstream analysis in VCaP cells. a** The upper bar plot shows the number of chimeric connections per kilobase across chromosomes, highlighting higher chimeric activity in Chr 19 and Chr M. The lower heatmap visualizes interchromosomal connections, with intensity indicating the count of connections between different chromosomes. **b** The bar plot shows the number of transcripts (in thousands) across different isoform classification categories. DeepChopper-processed reads result in a higher number of transcripts compared to Dorado-trimmed reads. The inset details the isoform classification scheme. **c** Detected gene fusions from Dorado adapter-trimmed reads and DeepChopper-processed reads. Gene fusions identified from short-read RNA-seq were used to validate fusion events detected

from dRNA-seq. **d** GO enrichment analysis of chimera artifact-affected genes, with color indicating gene count per term. Statistical significance assessed by one-sided Fisher's exact test (EASE score) with Benjamini-Hochberg FDR correction (FDR <0.05). **e** Analysis of a chimeric read (Read ID: 3b2292e9-43e5-4e40-87d9-ccc23897377c) artifact detected as an *RPS29-COX8A* fusion. The schematic shows the fusion between *RPS29* (Chr 14) and *COX8A* (Chr 11). The green plot indicates quality scores along the read, and the blue plot shows raw signal intensity (in pA). The chopped region identified by DeepChopper corresponds to a low-quality segment with low current intensity, polyA, and short open pore signals, suggesting the presence of an ONT adapter.

HepG2: 1.25%), representing 15,690–57,122 affected reads per sample (Supplementary Table 4). Critically, internal adapters accounted for 63–85% of all chimeric reads across cell lines, identifying adapter-bridged artifacts as the predominant source of false RNA rearrangement[22]. This systematic occurrence of internal adapters across all tested cell lines indicates that adapter-bridged chimeras

are not specific to VCaP but represent a general characteristic of dRNA-seq data. Across these samples, DeepChopper reduced unsupported chimeric alignments by 62% to 84%, while preserving cDNA-supported chimeric alignments without noticeable reduction (Fig. 2g). These results reinforce the widespread presence of chimera artifacts in dRNA-seq and the effectiveness of DeepChopper in

selectively removing them without compromising true biological signals.

We next applied DeepChopper to the human WTC11 induced pluripotent stem cell line using data from the Long-read RNA-Seq Genome Annotation Assessment Project (LRGASP)[23]. This dataset includes cDNA-based long-read sequencing generated with multiple protocols (PCR-cDNA, CapTrap, R2C2) across ONT and Pacific Biosciences (PacBio) platforms, providing a robust benchmarking resource. DeepChopper selectively eliminated only those chimeric alignments not supported by any cDNA-based method (Fig. 2h), further demonstrating its precision in distinguishing genuine chimeras from technical artifacts.

To assess cross-species generalizability, we extended the analysis to the F121-9 mouse embryonic stem cell line, also from the LRGASP dataset. DeepChopper reliably removed chimeric reads not supported by orthogonal cDNA-based sequencing platform (Supplementary Fig. 8), confirming its applicability to both human and non-human transcriptomes.

Importantly, across all datasets, DeepChopper consistently outperformed ONT's Dorado adapter trimming alone, even when applied as a post-processing step, underscoring its distinct and additive utility in chimera artifact correction.

## Chimera artifact analysis in RNA004 chemistry

Recently, ONT released a new SQK-RNA004 chemistry for dRNA-seq, but it remains unclear whether chimera artifacts persist with this update. To investigate, we generated new data from the VCaP cell line using this updated chemistry. We first applied DeepChopper in a zero-shot setting to assess cross-chemistry generalization, as the model was trained exclusively on RNA002 adapter patterns.

In zero-shot application, DeepChopper detected internal adapters in 0.33% of VCaP RNA004 reads (38,878 reads out of 11,714,520 total), lower than the 1.62% observed in VCaP RNA002 (Supplementary Table 4). DeepChopper reduced chimeric alignments by 21% compared to Dorado base-called and adapter-trimmed reads, increasing the proportion of cDNA-supported chimeric alignments from approximately 25% to 30% (Supplementary Fig. 9a). Similar results were observed when DeepChopper was applied after Dorado's adapter trimming, demonstrating compatibility with standard preprocessing pipelines. Internal adapter-like sequences identified by DeepChopper exhibited low base quality scores (mean Q-score: 7.85) and poor alignment identity to the human genome (mean BLAT identity: 0.38), supporting their classification as artifacts (Supplementary Fig. 9b).

To optimize performance on RNA004 data, we fine-tuned DeepChopper using a dataset with 300,000 reads created from VCaP RNA004 reads (70% training, 20% validation, 10% test) (See Methods for details). The fine-tuned model achieved marginal additional improvement, reducing chimeric alignments by 23–25% compared to Dorado-processed reads, a 3-4% improvement over the zero-shot model (Supplementary Fig. 10). Critically, both the original RNA002-trained model and the RNA004-fine-tuned model preserved all cDNA-supported chimeric alignments, demonstrating that DeepChopper specifically targets adapter-bridged artifacts rather than biological RNA rearrangements[22].

While DeepChopper's reduction of chimeric alignments in RNA004 (21-22%) is lower than in RNA002 (91%), both the reduced artifact prevalence (0.33% vs 1.62%) and the lower reduction magnitude are expected given chemistry improvements designed to reduce artifact formation[3,24]. Nonetheless, the systematic detection of internal adapters across both RNA002 and RNA004 chemistries confirms that adapter-bridged chimeras remain an inherent characteristic of current dRNA-seq workflows, and DeepChopper's ability to generalize across chemistries without retraining highlights its robustness for emerging platforms.

Both the original RNA002-trained model and the RNA004-fine-tuned model are available in the DeepChopper repository, providing users with optimized options for different sequencing chemistries.

## Impact on downstream transcriptome analysis

To investigate factors contributing to chimera artifact formation, we examined gene expression levels and transcript lengths associated with chimeric read artifacts. Genes involved in these artifacts showed higher expression than the general transcriptome ($p$ - value $<2.2 \times 10^{-16}$; Supplementary Fig. 11a), while exhibiting a similar gene length distribution (Supplementary Fig. 11b). Analysis of chimeric junctions across the genome revealed uneven distribution among chromosomes, with the mitochondrial chromosome (Chr M) showing the highest frequency of chimeric connections per base pair—suggesting a potential hotspot for artifact formation (Fig. 3a). This pattern persisted in RNA004 dRNA-seq data (Supplementary Fig. 11c), indicating that chimera artifacts remain a fundamental limitation of dRNA-seq, regardless of chemistry improvements.

We next assessed how DeepChopper correction influences downstream transcriptome analyses. Using IsoQuant[25] to annotate transcripts from VCaP dRNA-seq data, we found that DeepChopper nearly doubled the number of identified transcripts compared to uncorrected data (Fig. 3b). Similar results were observed with RNA004 data (Supplementary Fig. 11d) and when DeepChopper was applied after Dorado's adapter trimming. The largest gains were observed in full-length transcripts (Full splice match (FSM) category), with additional increases in alternatively spliced isoforms (Incomplete splice match (ISM), Novel in catalog (NIC), and Novel not in catalog (NNC) categories). These findings underscore the effectiveness of DeepChopper in mitigating the detrimental effects of chimera artifacts on transcript annotation.

To further assess the implications of artifact removal, we examined gene fusion detection. DeepChopper-corrected reads yielded an 89% reduction in gene fusion calls by FusionSeeker[26] compared to Dorado-trimmed data. Importantly, these reduced fusion calls were not supported by fusions detected in matched short-read RNA-seq data using Arriba[27] (Fig. 3c), suggesting they were false positives. Applying DeepChopper after Dorado trimming yielded consistent results, reinforcing its utility regardless of prior processing steps.

Closer inspection of the filtered gene fusion calls revealed a strong enrichment for ribosomal protein genes (Supplementary Fig. 12a). Gene Ontology (GO) enrichment analyses in VCaP (Fig. 3d) and SG-NEx cell lines confirmed this trend, with ribosomal genes frequently appearing in artifact-associated fusions. This enrichment extended to chimera artifacts in RNA004 data as well (Supplementary Fig. 12b). A manual review of a chimeric read identified as an *RPS29-COX8A* fusion revealed that the DeepChopper-processed region, interpreted as an internal adapter sequence, aligned with low-intensity raw current signals, which is consistent with ONT adapter characteristics (Fig. 3e). The presence of polyA and open pore signals at the boundary of this region further supported an artifact origin rather than a bona fide fusion event. Supplementary Fig. 13 illustrates the corresponding sequence and base qualities, including the alignment of upstream and downstream segments around the detected adapter, confirming that the apparent *RPS29-COX8A* fusion arises from an adapter-bridged chimera.

In summary, DeepChopper improves the quality of nanopore dRNA-seq data by accurately identifying and splitting adapter-bridged chimera that otherwise confound transcript annotation and gene fusion detection. These improvements are robust across different sequencing chemistries and preprocessing pipelines.

## Discussion

DeepChopper addresses a critical gap in ONT dRNA-seq: detecting and recovering adapter-bridged chimeric artifacts through internal

adapter identification. Our validation across multiple cell lines, species, and chemistries demonstrates internal adapters occur systematically in 0.33–1.62% of reads (tens to hundreds of thousands per experiment), with each propagating errors to transcript annotation, gene fusion detection, and expression quantification[22]. No prior literature has systematically characterized adapter-bridged chimera formation mechanisms in dRNA-seq; our work addresses this gap while providing the computational solution. DeepChopper identifies internal adapter sequences marking artificial junctions, then splits chimeras at these boundaries with single-nucleotide precision to recover component sub-reads. This adapter-based approach differs fundamentally from filtering methods that discard problematic reads. The recovery capability depends on detectable internal adapters, distinguishing adapter-bridged artifacts from RT-mediated template-switching in cDNA-seq, which creates chimeras without internal adapters[28,29].

DeepChopper leverages GLM capabilities unavailable to conventional approaches: (1) Long-range context[11] enabling full-transcript scanning (median ~2.7 kb, 99th percentile ~14 kb); (2) Transfer learning enabling robust detection despite sequencing errors; (3) Quality-aware predictions integrating per-base confidence (F1: $0.97 \rightarrow 0.99$); (4) Alignment-free operation eliminating mapping biases.

Performance differences between DeepChopper and existing tools reflect capability gaps, not quality. Each tool excels at its intended application: Pychopper for cDNA primer detection[7,30] and Dorado for 3′ adapter trimming[5]. However, none detect internal adapters in dRNA-seq. Pychopper targets cDNA-specific primers absent in dRNA-seq. Porechop/Porechop_ABI require exact matching or stable k-mer patterns[8,9], an approach incompatible with unknown, corrupted adapters mis-called by RNA basecallers. Negligible performance on internal detection confirms this is a genuinely unsolved problem.

Alignment-based methods employ ruled-based filtering, flagging aberrant alignment patterns, rather than detecting root causes. Breakinator[18] detects RT-mediated foldbacks in cDNA-seq using distance rules; its validation confirms failure in dRNA-seq where RT artifacts don't occur. FLAIR-fusion[31] is a specialized fusion caller, not general quality control. Distance rules cannot distinguish adapter-bridged artifacts from biological RNA rearrangements[22] and they cannot handle artifacts that do not map as chimeric reads or that fail to map at all. In contrast, internal adapter presence definitively indicates artifacts, enabling selective correction while preserving biological complexity.

Validation across RNA002 and RNA004 with Dorado basecalling shows adapter-bridged chimeras persist systematically despite chemistry improvements[3,24]. RNA004 fine-tuning achieved 3-4% additional improvement beyond zero-shot RNA002 model performance, with both models preserving cDNA-supported events. This demonstrates learned probabilistic patterns enable robust generalization without protocol-specific recalibration, which is critical as technologies evolve. Challenging scenarios (Supplementary Fig. 14, Supplementary Fig. 15) include incomplete 3′ end detection in multi-adapter reads and partial detection of degraded sequences. Solutions include combining Dorado (3′ end adapters) with DeepChopper (internal adapters) (Supplementary Fig. 16) and post-processing length filtering (default 20 bp minimum). Despite edge cases, DeepChopper achieves 62–91% artifact reduction while preserving cDNA-supported biological events.

DeepChopper demonstrates how GLM address long-read sequencing challenges resisting conventional approaches. Long-range context, error-tolerant learning, single-nucleotide precision, and quality-aware predictions enable detecting corrupted internal adapters that elude exact-matching, k-mer, or alignment methods. Recovery rather than removal represents a paradigm shift from filtering to correction, preserving valuable data while enhancing accuracy. Future directions include extended context for ultra-long transcripts, alternative approaches for RT-mediated cDNA-seq chimeras lacking internal adapters, and expansion to additional platforms. This advance enables confident biological interpretation of dRNA-seq data (from isoform discovery to gene fusion detection), strengthening transcriptomic research in complex biological systems where accurate transcript characterization is essential for understanding gene regulation and cellular function.

## Methods

### Cell culture

The human male prostate cancer cell line VCaP was obtained from the American Type Culture Collection (ATCC, Cat# CRL-2876) and cultured under sterile conditions to maintain optimal growth and viability. The cells were grown in Dulbecco's Modified Eagle Medium (DMEM, high glucose; Gibco, Cat# 11-965-092) supplemented with 10% fetal bovine serum (FBS Opti-Gold, Performance Enhanced, US Origin; Gendepot, Cat# F0900-050) to provide essential growth factors. In addition, the culture medium was enriched with 5 ml of 100 mM Sodium Pyruvate (Gendepot, Cat# CA017-010) to support cellular metabolism and 5 ml of Antibiotics-Antimycotics (100 × ) (Gendepot, Cat# CA002-010) to prevent microbial contamination. Cells were cultured in 100mm cell culture treated dishes (Thermo Fisher Scientific, Cat# 12-556-002) and incubated at 37 °C in a humidified atmosphere containing 5% $CO_2$, with media changes performed every 72 h to ensure nutrient availability and waste removal. Cell confluency was regularly monitored and subculturing was performed before reaching 80% confluency to maintain healthy growth conditions and prevent over-confluence stress.

### RNA extraction and quantification

Total RNA was extracted using the RNeasy Mini Kit (Qiagen, Cat# 74104) according to the protocol of the manufacturer. The quality and concentration of RNA were assessed using an Agilent 2100 Bioanalyzer. Poly(A)+ RNA was then enriched from total RNA using the Dynabeads™ mRNA Purification Kit (Invitrogen, Cat# 65001), which utilizes oligo (dT) beads for selective mRNA binding. The mRNA was quantified using a Qubit 4 fluorometer and a Qubit RNA HS Assay Kit (Thermo Fisher Scientific, Cat# Q32852). The mRNA preparations were either immediately used to prepare a sequencing library or frozen and stored at -80 °C until further use.

### Nanopore sequencing

We performed nanopore dRNA-seq sequencing of the enriched mRNA using two different sets: the RNA002 kits with R9.4.1 flow cells and the RNA004 kits with R10.4.1 flow cells. The decision to incorporate the RNA004 kit, a newly released option, was driven by our intention to test its capabilities in conjunction with our DeepChopper tool to optimize data quality and sequencing efficiency. For the RNA002 library, 1 μg of poly(A)+ RNA was used as input for library preparation using the Direct RNA Sequencing Kit (SQK-RNA002, ONT) following the manufacturer's instructions. Nanopore dRNA-seq employs a reverse transcriptase adapter (RTA) that typically binds to the poly(A) tails of messenger RNA (mRNA); subsequently, a sequencing adapter is ligated to the RTA, which guides the mRNA through the nanopore for sequencing. The prepared library was loaded onto four MinION R9.4 flow cells (FLO-MIN106) and sequenced for 48 h using the Oxford Nanopore MinION device. For the RNA004 library, 300 ng of poly(A)+ RNA was used as input for library preparation using the Direct RNA Sequencing Kit (SQK-RNA004, ONT) according to the protocol of the manufacturer. The library was then loaded onto a PromethION RNA Flow Cell (FLO-PRO004RA) and sequenced on the Oxford Nanopore PromethION device for 72 h.

For Direct cDNA sequencing, we utilized the Direct cDNA Sequencing Kit (SQK-DCS109, ONT) following the manufacturer's protocol. Briefly, 5 μg of total RNA was used as input for first-strand cDNA synthesis using Maxima H Minus Reverse Transcriptase (Thermo Fisher Scientific) with the SSP and VN primers provided in the kit. To

eliminate potential RNA contamination, we treated the sample with RNase Cocktail Enzyme Mix (Thermo Fisher Scientific). Second-strand cDNA synthesis was carried out using LongAmp Taq Master Mix (New England Biolabs). The resulting double-stranded cDNA underwent end-repair and dA-tailing using the NEBNext Ultra End Repair/dA-Tailing Module (New England Biolabs). Subsequently, sequencing adapters were ligated to the prepared cDNA using Blunt/TA Ligase Master Mix (New England Biolabs). Between each enzymatic step, the cDNA and libraries were purified using AMPure XP beads (Agencourt, Beckman Coulter). We quantified the libraries using a Qubit Fluorometer 3.0 (Life Technologies) to ensure adequate concentration and quality. The final library was loaded onto a MinION R9.4 flow cell and sequenced on the Oxford Nanopore MinION device for 72 h.

### Training data preparation

We acquired ONT dRNA-seq FAST5 data from the SG-NEx project, which includes six human cell lines: HEK293T, A549, K562, HepG2, MCF7, and HCT116[17]. The FAST5 files were converted to POD5 format using the POD5 conversion tool (https://pod5-file-format.readthedocs.io). Subsequently, FASTQ files were generated using Dorado (v0.5.2)[5] with adapter trimming disabled (`-no-trim`) and the rna002_70bps_hac@v3 model. The reads were then aligned to the human reference genome (GRCh38) using minimap2 (v2.24)[20] with ONT direct RNA-specific parameters (-ax splice -uf -k14) for optimized alignment. The resulting SAM files were then converted to BAM format, indexed, and sorted using SAMtools (v1.19.2)[32].

For adapter sequence extraction, we selected primary alignments without supplementary alignments and implemented a refined identification protocol. While 3′ end soft-clipped regions were candidates for adapter sequences, we did not assume all such regions corresponded to adapters. Instead, we incorporated a critical biological refinement step: we first identified polyA tails at the beginning of soft-clipped regions, as these represent reliable biological indicators of transcript termination. Only sequences following these polyA tails were designated as potential adapter sequences, while aligned regions were classified as non-adapter sequences. This approach improved the precision of our training data by distinguishing true adapter sequences from other non-adapter soft-clipped regions that might result from alignment artifacts or sequencing errors. By anchoring our adapter identification to known biological features, we reduced the risk of misclassification and ensured the training data more accurately reflected the natural transcript-adapter boundaries encountered in dRNA-seq.

To create artificial chimeric reads, we randomly combined two non-adapter sequences with one adapter sequence to create FASTQ records. The dataset consists of positive examples containing adapter sequences (with a 1:1 ratio of 3′ end and internal adapters) and negative examples without any adapter sequences, in a 9:1 ratio. In total, 600,000 data points were generated and divided into training (N = 480, 000), validation (N = 60, 000), and test sets (N = 60, 000) in an 8:1:1 ratio using stratified random sampling.

### Language model architecture

DeepChopper approaches adapter sequence identification as a token classification task, utilizing a model with 4.6 million trainable parameters. The system tokenizes biological sequences at single-nucleotide resolution, with each nucleotide (A, C, G, T, and N) serving as a fundamental token. This nucleotide-level granularity enables precise discrimination between artificial adapter sequences and native biological sequences.

At its core, DeepChopper employs HyenaDNA[11] as its primary feature extractor. HyenaDNA processes the input sequence using multiple attention-free linear layers with a receptive field, transforming the nucleotide tokens into rich 256-dimensional feature representations. The model handles variable-length sequences

through a padding approach, maintaining consistent performance across different sequence lengths while efficiently capturing long-range dependencies.

These features are then fed through a quality block that incorporates standardized base quality scores. Prior to processing, the quality scores are normalized using z-score standardization ($\mu = 0$, $\sigma = 1$) to ensure numerical stability. The quality block, comprising two MLPs with residual connections (hidden dimensions: 256), processes this normalized quality information while preserving the original sequence features. Each MLP layer is followed by ReLU activation, enhancing the model's ability to learn complex quality-sequence relationships.

The processed sequence features are subsequently fed into a classification head consisting of a two-layer neural network. This architecture transforms the feature representations into classification outputs at the nucleotide level. The classification module employs a softmax activation function to compute probability distributions across two classes: adapter and non-adapter. For a given nucleotide position with output logits $z_1$ and $z_2$ (corresponding to adapter and non-adapter classes), the softmax function computes class probabilities as:

$$P(y_i = c) = \frac{e^{z_c}}{\sum_{j=1}^{2} e^{z_j}} \tag{1}$$

where $P(y_i = c)$ represents the probability that nucleotide position $i$ belongs to class $c$. The final classification decision is based on the class with the higher probability score. In other words, a nucleotide is classified as an adapter if $P(y_i = \text{adapter}) > P(y_i = \text{non-adapter})$. Hence, a threshold of 0.5 is applied implicitly.

This nucleotide-level classification strategy allows DeepChopper to identify adapter boundaries with high precision, including both terminal and internal adapter sequences.

### Model training

DeepChopper processes sequences up to 32,770 nucleotides in length, excluding any longer sequences from analysis. To ensure efficient batch processing, shorter sequences were padded to this maximum length. The model was trained using a supervised learning approach, utilizing sequences labeled with adapter annotations. Training was performed in a High Performance Computing (HPC) cluster using two A100 Graphics Processing Units (GPUs). The batch size was set to 64, and validation was performed every 20,000 steps. The model with the highest validation F1 score for the base prediction task was selected for subsequent analyses. Training was carried out over 60 epochs, with early stopping applied based on validation performance to mitigate overfitting risks.

The Adam optimizer was used for parameter optimization, with settings of $\beta_1 = 0.9$ and $\beta_2 = 0.999$[33]. A learning rate scheduler was used to reduce the learning rate when validation loss ceased improving, starting with an initial learning rate of $2 \times 10^{-5}$. The cross-entropy loss function was used to update the model parameters, defined as follows:

$$\mathcal{L}_{\text{BCE}} = -\frac{1}{N} \sum_{i=1}^{N} [y_i \log(\widehat{y_i}) + (1 - y_i) \log(1 - \widehat{y_i})] \tag{2}$$

where $\mathcal{L}_{\text{BCE}}$ is the binary cross-entropy loss, $N$ is the total number of tokens in the input sequence, $y_i$ is the ground true label for the $i$-th token, and $\widehat{y_i}$ is the predicted probability for the $i$-th token.

The average cross-entropy loss across the mini-batch is computed as:

$$\mathcal{L}_{\text{BatchBCE}} = \frac{1}{B} \sum_{j=1}^{B} \mathcal{L}_{\text{BCE}}(\mathbf{y}_j, \widehat{\mathbf{y}}_j) \tag{3}$$

where $\mathcal{L}_{BatchBCE}$ is the average binary cross-entropy loss for the mini-batch, $B$ is the batch size (number of sequences in the mini-batch), and $\mathbf{y}_j$ and $\hat{\mathbf{y}}_j$ are the true labels and predicted probabilities for the $j$-th sequence in the batch.

The model evaluation metrics included accuracy, precision, recall and the F1 score, calculated using the following equations:

$$\text{Precision} = \frac{TP}{TP + FP} \quad (4)$$

$$\text{Recall} = \frac{TP}{TP + FN} \quad (5)$$

$$F1 = 2 \times \frac{\text{Precision} \times \text{Recall}}{\text{Precision} + \text{Recall}} \quad (6)$$

The final selection of the model was based on the optimal performance in the validation set. The model is implemented by PyTorch (v2.5.0)[34]. To identify the best hyperparameter configuration, the Hydra (v1.3.2)[35] framework was used.

## Ablation study of quality block component

An ablation study was conducted to compare two model variants: one with the quality block component and one without. Both models were trained using the same dataset of 100,000 samples, following the training procedures described in Training data preparation and Model training. All hyperparameters, including learning rate, batch size, and optimization algorithm, were kept constant across both configurations. The only architectural difference was the inclusion or exclusion of the quality block. Evaluation was performed on a held-out test set using the F1 score as the primary metric.

## RNA004 chemistry fine-tuning

To optimize DeepChopper's performance for the current RNA004 sequencing chemistry, we fine-tuned the model using VCaP RNA004 data. Following the same data preparation methodology described in Training data preparation, we generated synthetic training data from VCaP RNA004 base-called reads. The fine-tuning dataset consisted of 300,000 reads, split into training (70%, 210,000 reads), validation (20%, 60,000 reads), and test (10%, 30,000 reads) sets, with positive examples containing adapter sequences (1:1 ratio of 3′ end and internal adapters) and negative examples without adapters in a 9:1 ratio.

The fine-tuning process maintained the same model architecture and hyperparameters as the original RNA002 training but allowed the model to adapt to RNA004-specific error profiles and signal characteristics. Training was performed on two NVIDIA A100 GPUs using the optimization settings described in Model training, with early stopping based on validation F1 score. Fine-tuning required approximately 9 hours of total training time. Performance evaluation compared three conditions: (1) Dorado basecalling with adapter trimming (baseline), (2) Dorado followed by the original RNA002-trained DeepChopper, and (3) Dorado followed by the RNA004-fine-tuned DeepChopper. Chimeric alignments were evaluated against matched cDNA sequencing data to distinguish biological events from artifacts.

## Sliding window approach for prediction refinement

To improve prediction consistency and reduce local noise, a sliding window approach was implemented for post-processing of nucleotide-level classification outputs. This method extends predicted adapter regions and smooths isolated predictions, better reflecting the typical length distribution of adapter sequences in ONT dRNA-seq data. The approach enhances continuity in adapter-labeled regions and mitigates the occurrence of fragmented or spurious classifications.

The refinement operates on the initial (raw) predictions from the model rather than iteratively refining predictions at each step, a design choice driven by the requirement for precise adapter boundary detection at single-nucleotide resolution. For each base position $i$, we define a window $W_i$ of size $w$ (default: 21 bp) centered at position $i$. The final classification $y_i$ is determined by majority vote of all predictions within the window:

$$y_i = \begin{cases} 1 & \text{if } \sum_{j=i-k}^{i+k} p_j > \frac{W}{2} \\ 0 & \text{otherwise} \end{cases} \quad (7)$$

where $y_i$ is the final prediction for the $i$-th nucleotide, $W$ is the sliding window size, $k$ is half the window size ($k = \frac{W-1}{2}$), and $p_j$ represents the initial predicted label for the $j$-th nucleotide within the window, where a value of 1 indicates that the nucleotide is part of an adapter sequence, and a value of 0 indicates that it is part of a non-adapter sequence.

The default window size is set to 21 nucleotides, and can be customized using the –smooth-window parameter in the DeepChopper implementation to accommodate dataset-specific characteristics.

## Post-processing and filtering

After refining the adapter predictions, four filtering steps were applied to enhance the quality of the final results:

1. A predicted adapter sequence must be at least 13 nucleotides long. Sequences shorter than this length threshold are not considered valid adapters.
2. If a read contains more than four adapter sequences, the entire read sequence is retained without any adapter removal.
3. For reads containing four or fewer adapter sequences, the identified adapters are removed and the read is divided into smaller segments.
4. Any segments resulting from this process that are less than 20 nucleotides long are discarded.

Each remaining segment and its corresponding base quality scores are stored as a single read record in the final FASTQ file. This filtering process separates chimeric read artifacts containing internal adapters into multiple segments, while retaining reads with 3′ end adapters as single shortened segments.

All filtering thresholds, including minimum segment length, and maximum adapter count per read, are configurable via command-line parameters in the DeepChopper implementation, allowing users to tailor these settings to dataset-specific requirements or experimental conditions.

## Computational benchmarks

All benchmarks were conducted in triplicate using btop (https://github.com/aristocratos/btop, v1.4.0) and nvtop (https://github.com/Syllo/nvtop, v3.1.0) to monitor CPU and GPU memory usage, respectively. Evaluations were performed on high-performance computing infrastructure with 16 CPU cores, 60 GB RAM, and dual NVIDIA A100 GPUs (80 GB memory each). Adapter prediction stage used a batch size of 64.

## Transcript length distribution analysis

To assess the biological appropriateness of DeepChopper's 32 kb input limit, we analyzed transcript length distributions from two sources: theoretical annotations and empirical sequencing data. For theoretical analysis, all protein-coding transcripts were extracted from Ensembl human genome annotations (GRCh38.115, released July 11, 2025). We calculated median length, 95th percentile, 99th percentile, and the fraction of transcripts exceeding 32 kb using Python. For empirical analysis, read length distributions were examined across seven dRNA-seq datasets: A549, MCF7, HCT116, K562, HepG2, VCaP RNA002, and VCaP RNA004. For each dataset, we calculated comprehensive

statistics including minimum, maximum, mean, standard deviation, quartiles (Q1, Q2, Q3), and high percentiles (P90, P95, P99). The number and percentage of reads exceeding the 32 kb threshold were specifically quantified to assess the practical impact of this limitation. All read length statistics were computed from Dorado (v0.5.2) base-called reads with the trim option enabled.

## Validation of chimera artifact reduction

Cross-platform validation of dRNA-seq chimera artifacts identified by DeepChopper was conducted leveraging ONT direct cDNA sequencing and additional cDNA-based sequencing platforms. Direct cDNA sequencing validation was performed using six cancer cell lines, including the VCaP dataset generated in this study and five published datasets (A549, K562, HepG2, MCF7, and HCT116) obtained from the SG-NEx project[17]. The direct cDNA data in FAST5 format were converted to POD5 format using the POD5 conversion tool (https://pod5-file-format.readthedocs.io). Subsequently, FASTQ files were generated using Dorado (v0.5.2)[5] with adapter trimming enabled (–trim adapters) and the dna_r9.4.1_e8_hac@v3.3 model. The reads were then processed using Pychopper (https://github.com/epi2me-labs/pychopper, v2.7.9) and Cutadapt (v4.2)[36] according to a published protocol[37]. The oriented reads were aligned to the human reference genome (GRCh38) using minimap2 (v2.24)[20] with optimized parameters (-ax splice -uf -k14) for spliced alignment. The resulting SAM files were then converted to BAM format, indexed, and sorted using SAMtools (v1.19.2)[32].

Additional cDNA-based long-read sequencing data from the WTC11 (human) and F121-9 (mouse) cell lines were used for further validation, incorporating five distinct platforms: ONT PCR-cDNA, ONT CapTrap, ONT R2C2, PacBio cDNA, and PacBio CapTrap. The raw FASTQ files (and FASTA files for ONT R2C2) from these datasets were provided by the LRGASP project[23]. For the PCR-cDNA data, the reads were processed using Pychopper (https://github.com/epi2me-labs/pychopper, v2.7.9) and Cutadapt (v4.2)[36], following the protocol described in ref.[37]. ONT reads were then aligned to the human reference genome (GRCh38) or mouse reference genome (GRCm39) using minimap2 (v2.24)[20] with the parameters (-ax splice -uf -k14), while PacBio reads were aligned using the parameters (-ax splice:hq -uf). The ONT dRNA-seq data from A549, K562, HepG2, MCF7, HCT116, VCaP, WTC11 and F121-9 cell lines were processed using the same pipeline, except that the F121-9 cell line data was aligned to the mouse reference genome (GRCm39).

To validate the chimeric alignments derived from dRNA-seq, comparisons were made with chimeric alignments identified from cDNA-based data across the specified platforms. Chimeric alignments, defined by a primary alignment and one or more supplementary alignments, each containing the SA tag in the BAM file, were converted into lists of genomic intervals based on their corresponding alignments. The genomic interval lists were then compared between platforms, and overlapping intervals were considered concordant if the distance difference between them was less than 1000 bp. Supporting rates were calculated as the proportion of dRNA-seq chimeric alignments corroborated by cDNA-based platforms, thereby providing cross-validation of chimera artifacts identified by DeepChopper.

To empirically assess the applicability of alignment-based artifact detection methods to dRNA-seq, we compared DeepChopper with Breakinator (v1.0)[18] using VCaP RNA002 data. Three processing pipelines were evaluated: Dorado basecalling with trim option (baseline), Dorado with trim followed by DeepChopper, and Dorado with trim followed by Breakinator. All pipelines started with identical Dorado-trimmed reads to enable direct comparison. Breakinator was applied using default parameters as specified in its documentation[18]. Chimeric alignments from each pipeline were classified as cDNA-supported (validated by matched direct cDNA sequencing) or unsupported (likely artifacts) using the validation framework described above. Support

rates were calculated as the proportion of chimeric alignments corroborated by cDNA-seq data.

## Internal adapter sequence characterization

To assess the quality and genomic origin of internal adapter sequences, we randomly subsampled 1 million reads from the VCaP RNA002 and VCaP RNA004 datasets. DeepChopper was applied to identify internal adapter sequences within these reads. For base quality analysis, we extracted quality scores for all nucleotide positions identified as internal adapters, yielding 146,580 sequences from RNA002 and 11,143 from RNA004. Base quality scores (Phred Q-scores) were obtained directly from FASTQ files. For genome alignment analysis, internal adapter sequences were aligned to the human reference genome (GRCh38) with default parameters. Sequences that failed to align to the genome were excluded from the identity distribution analysis. This resulted in 98,807 sequences from RNA002 and 6185 sequences from RNA004 being included in the genome identity analysis. Sequences that failed to align to the genome were excluded from the identity distribution analysis, as they would yield uninformative zero identity scores.

## BLAT identity calculation

The accuracy of DeepChopper in detecting adapter sequences was evaluated by aligning the identified sequences to the human reference genome using BLAT[21]. A BLAT identity score was defined as the ratio of matched bases to the total sequence length:

$$\text{BLAT Identity} = \frac{\text{Match Length}}{\text{Sequence Length}} \tag{8}$$

In this context, match length refers to the number of bases in the query sequence that align with the reference genome, while sequence length denotes the total length of the query sequence. This score provides a quantitative measure of how closely each identified sequence aligns with the reference genome, serving as an indicator of detection accuracy. The alignments were performed using the PxBLAT (v1.2.1)[38]

## Gene expression analysis and transcript classification

Gene expression levels from dRNA-seq were quantified using IsoQuant (v3.1.2)[25], with the parameters (–data_type nanopore –stranded forward –model_construction_strategy default_ont –sqanti_output). The –sqanti_output option enables IsoQuant to generate files containing transcript classification information, analogous to the output provided by SQANTI[39].

## Gene fusion identification and visualization

For ONT dRNA-seq data, gene fusions were identified using FusionSeeker (v1.0.1)[26] with default settings. For short-read RNA-seq data, FASTQ files for the VCaP cell line were obtained from the Cancer Cell Line Encyclopedia (CCLE) project[40] under SRA accession SRX5417211. Raw reads were mapped to the hg38 reference genome using STAR (v2.7.11)[41], and gene fusion events were detected with Arriba (v2.4.0)[27]. The gene structure of the *RPS29-COX8A* fusion was visualized using GSDS (v2.0)[42]. Base quality scores were generated with a custom Python script, and ion current signals were visualized using Squigualiser (v0.6.3)[43]. The circos plot for gene fusion events was visualized using chimeraviz (v1.30.0)[44].

## GO enrichment analysis

GO enrichment analysis of biological processes for genes involved in chimera artifacts identified in dRNA-seq data was performed using the Database for Annotation, Visualization, and Integrated Discovery (DAVID) webserver[45]. Enrichment significance was assessed using a

one-sided modified Fisher's exact test (EASE score). P values were adjusted for multiple comparisons using the Benjamini-Hochberg false discovery rate (FDR) method. Terms with FDR <0.05 were considered significant.

## Computing resource

All computations were performed on a HPC server equipped with a 64-core Intel(R) Xeon(R) Gold 6338 CPU and 256 GB of RAM. The server was also configured with two NVIDIA A100 GPUs, each with 80 GB of memory, enabling efficient processing of both CPU-intensive tasks and GPU-accelerated deep learning workloads.

## Reporting summary

Further information on research design is available in the Nature Portfolio Reporting Summary linked to this article.

## Data availability

Raw and processed data generated in this study, including dRNA-seq using the SQK-RNA002 and SQK-RNA004 kits, as well as direct cDNA sequencing of VCaP cells, have been deposited in the Gene Expression Omnibus (GEO) under the accession number GSE277934. Source data are provided with this paper.

## Code availability

DeepChopper (v1.2.6), implemented in Rust and Python, is open source and available on GitHub (https://github.com/ylab-hi/DeepChopper) under the Apache License, Version 2.0. The package can be installed via PyPI (https://pypi.org/project/deepchopper/) using pip, with wheel distributions provided for Windows, Linux, and macOS to ensure easy cross-platform installation. An interactive demo is available on Hugging Face (https://huggingface.co/spaces/yangliz5/deepchopper), allowing users to test DeepChopper's functionality without local installation. For large-scale analyses, we recommend using DeepChopper on systems with GPU acceleration. Detailed system requirements and optimization guidelines are available in the repository's documentation.

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

## Acknowledgements

This project was supported in part by NIH grants R35GM142441 and R01CA259388 awarded to RY, and NIH grants R01CA256741, R01CA278832, and R01CA285684 awarded to Q. C.

## Author contributions

Y.L., T.Y.W. and R.Y. designed the study with Q.C. Y.L. and T.Y.W. performed the analysis. Q.G., Y.R. and X.L. performed the experiments. YL designed and implemented the model and computational tool. Y.L., T.Y.W., Q.G. and R.Y. wrote the manuscript. R.Y. supervised this work.

## Competing interests

R.Y. has served as an advisor/consultant for Tempus AI, Inc. This relationship is unrelated to and did not influence the research presented in this study. Other authors have no competing interests.
