## [Transparent Peer Review file · Nature Communications]

Genomic Language Model Mitigates Chimera Artifacts in Nanopore Direct RNA Sequencing

Corresponding Author: Dr Rendong Yang

Version 0:

Reviewer comments:

Reviewer #1

(Remarks to the Author)

This manuscript introduces DeepChopper, a genomic language model specifically designed to mitigate chimera artifacts in nanopore direct RNA sequencing. The authors leverage HyenaDNA, an efficient genome language model utilizing long-range implicit convolution, to generate output embeddings. This is coupled with a multi-layer perceptron (MLP) and a sliding window-based probability prediction mechanism, enhanced by majority voting for robust predictions. As the first genomic language model tailored for long-read sequencing data, this work marks a significant advancement in transcriptomics. It addresses a critical challenge in the field, showcasing its utility in improving data quality for applications such as transcript annotation and gene fusion detection. The study is innovative, methodologically sound, and a valuable contribution to genomic research. Below, I highlight the manuscript's strengths and suggest revisions to further improve its clarity and impact.

Strength:

1. DeepChopper is the first genomic language model tailored for long-read sequencing data. It provides an end to end pipeline for chimera affected DNA sequence task. Its ability to handle sequences up to 32K nucleotides, combined with its integration of sequencing quality scores, ensures precise detection of adapter sequences at single-nucleotide resolution.
2. The tool is rigorously validated across multiple platforms, including ONT and PacBio, and different chemistries (RNA002 and RNA004). Its robustness and practical impact on reducing false positives in gene fusion detection are well-documented.
3. The method is comparatively faster than traditional methods due to Hyena's long range efficient capabilities and efficient computational complexity.
4. Improvements in transcript annotation and artifact reduction have strong implications for advancing transcriptomics research.

Suggestions for Revision:

1. Sliding Window Rationale: While the 21-nucleotide sliding window approach achieves an effective balance between smoothing and sensitivity, further explanation of how this parameter was optimized would strengthen the methodology section. Clarifying whether alternative window sizes were tested and how their performance compared would provide additional insights.
2. Other baseline models: Given that other Genome based language models exist (such as DNABERT, DNABERT2, The Nucleotide Transformers), authors might include results from these models as baselines or discussion of these models to better understand the capabilities of DeepChopper (in terms of accuracy).
3. Handling of Long Reads: The manuscript notes that sequences exceeding 32K nucleotides are excluded from analysis. Additional details on the prevalence of such sequences in datasets and the implications of this constraint would help users understand the tool's potential limitations.

4: Performance with RNA004 Chemistry: The reduced efficacy of DeepChopper with RNA004 compared to RNA002 (21% vs. 91% reduction in chimeric alignments) is an important observation. Including a discussion on differences in adapter designs or sequencing chemistry and their potential impact on performance would add depth to the analysis.

5. Computational Resource Documentation: Providing a table summarizing runtime, memory, and GPU/CPU requirements for datasets of varying sizes would be highly beneficial for researchers planning large-scale analyses.

6. Generalization Guidelines: Including recommendations for parameter tuning in diverse datasets or scenarios would enhance the tool's applicability across different research contexts.

(Remarks on code availability)

The reviewer did not run the code, but review the code and document provided in the GitHub. Github provide all the necessary documents to run the experiments. Overall, the open-source implementation and availability on platforms such as PyPI and Hugging Face ensure broad accessibility for the research community.

Reviewer #2

(Remarks to the Author)

This manuscript presents DeepChopper, a deep-learning-based tool designed to identify and remove chimera artifacts in Nanopore direct RNA sequencing (dRNA-seq). The authors propose that sequencing artifacts, particularly internal adapters, introduce errors that affect downstream analyses such as transcript annotation and gene fusion detection. DeepChopper leverages HyenaDNA, a genomic language model originally trained on DNA sequences, and integrates sequencing quality scores to improve adapter detection. The study evaluates DeepChopper using synthetic training data and real dRNA-seq datasets, claiming a significant reduction in chimeric reads compared to ONT's Dorado basecaller. However, several concerns arise regarding the justification, methodology, and evaluation of the proposed approach.

1. The manuscript does not clearly explain why a DNA-trained genomic language model (HyenaDNA) is appropriate for RNA sequencing data. RNA has different sequence structures, error profiles, and biological contexts than DNA. Since HyenaDNA was trained specifically on human DNA, it likely requires fine-tuning to adapt to RNA sequencing artifacts. The authors should provide a stronger theoretical justification for why this model is expected to work and discuss whether any modifications were made to improve its performance on RNA data. In addition, if non-human dRNA-seq data were to be analysed, DeepChopper, although not evaluated, will likely to be fail.

2. DeepChopper is introduced as a novel method, but the manuscript does not compare it to existing adapter trimming and chimera detection tools such as Porechop, Pychopper, or ONT's Dorado with adapter trimming enabled (rDNA-seq data was base-called with "--no-trim" option). Without direct comparisons in terms of accuracy, precision, recall, F1-score, and runtime, it is unclear whether DeepChopper offers meaningful advantages. A benchmarking analysis is necessary to determine whether DeepChopper outperforms or complements these existing methods.

3. The training data were created by manually inserting adapters into reads and identifying adapter regions using minimap2 soft-clipping. This process assumes that all 3' soft-clipped regions correspond to adapter sequences, which may not be true. Furthermore, chimeric reads were artificially generated, which does not necessarily reflect real ONT sequencing artifacts. The authors should validate their training data approach by analyzing real dRNA-seq datasets to determine whether internal adapters appear at similar frequencies and distributions. In addition, the authors used minimap2 (a long-read aligner) to generate true training data. Why don't use alignment-based method directly? How the genomic language model will be better than alignment-based methods? Basically, all the benchmarks (e.g. counting chimeric reads, etc.) were all based on alignment results. This needs to be justified.

4. DeepChopper classifies bases as adapters if the probability score is greater than 0.5, but the manuscript does not justify this threshold. If probability scores are clustered around 0.5, the classification may be unreliable. The authors should provide a distribution of probability scores and assess whether different cutoffs affect performance. An ROC curve or precision-recall analysis would help determine the optimal threshold for adapter detection.

5. The manuscript introduces a Quality Block to incorporate sequencing base quality scores, but it does not clearly explain how this improves model performance. Similarly, the sliding window refinement approach lacks detailed methodology, including how window size and majority voting criteria were selected. The authors should provide an ablation study testing the impact of the Quality Block and a sensitivity analysis for the sliding window parameters.

6. The manuscript states that DeepChopper has 4.6 million parameters, suggesting it is computationally efficient, but it does not provide any runtime or memory benchmarks. There is no information on how long DeepChopper takes to process a dataset, what GPU/CPU resources are required, or how it scales with large sequencing datasets. A computational efficiency evaluation is necessary to determine whether DeepChopper is practical for large-scale sequencing projects.

7. The validation dRNA-seq data were processed using Dorado with the “–no-trim” option, which preserves adapters and likely increases the observed chimeric read count. This choice could artificially inflate the problem DeepChopper aims to solve, making the model appear more effective than it actually is. The authors should repeat their analysis using Dorado with adapter trimming enabled to assess whether DeepChopper still provides a significant advantage.

(Remarks on code availability)

The format looks fine. However, I have no data to test it.

Version 1:

Reviewer comments:

Reviewer #1

(Remarks to the Author)

The authors have addressed all the comments and concerns I raised in the previous review. I recommend that the paper be published in its current form.

(Remarks on code availability)

Reviewer #3

(Remarks to the Author)

Summary:

The manuscript describes genomic language model based method for detecting chimeric reads in direct RNA sequencing. Overall, the manuscript is well written and thoughtfully organised, where the comments are also well addressed. I have a few questions remain after reading the manuscript, which either arise from reading the manuscript or following up to previous question which I felt more can be done to address those questions.

Major comments (more general):

The manuscript focuses on direct RNA seq, I wonder how generalizable is the approach to cDNA seq data, would a lot effort be needed? It would be good if authors could discuss for this application given there are a lot cases where Nanopore cDNA samples are generated instead.

The novelty of this approach, is the application of genomic language models, I agree. However, the advantage of this model vs conventional non-large language model is not discussed enough for why this approach should be favored. One point related is the testing data being too easy, which is shown as clear bimodal distribution as presented in response figure 6, any challenging scenarios that could benefit from deepchopper but not other methods?

(more specific)

Line 059-064: The chimera artifacts in dRNA-seq resulted from ligation (biases? I assume) or software missing open pore signal. The citation seems to describe more in a multiplexing context. I wonder is there any paper discussing this for regular ONT dRNA-seq instead? Also, the part software missing open pore signals is from paper in 2020, specifically referring to MinKNOW, which has been outdated, have the authors checked whether this is still a problem with the recent dRNA-seq?

Related to the handling of long-reads, as questioned by Reviewer 1, 32kb misses a few genes, for example SMAD2, LTRT, TTN (TTN actually is longer than 32kb?), etc? How are these genes dealt with then? And what impact will happen to these genes? In terms of expression levels? The read length referred to citation in the response is rather outdated, and the annotation with respect is also quite old.

Following Review 1 Q4 regarding RNA004, for practicality, it would be good if the authors could show fine-tuned results of deep-chopper for RNA004 in terms of performance to be more relevant in direct RNA sequencing context.

Following Reviewer 1 Q5 response, the authors demonstrated it in different data size, in terms of practicality, an data of size 20/25 million would also be interested to be included for this guidance.

Figure 1 c). Sliding window worth more explanation for better understanding. For example, from step 3 to step 4, the bottom 3rd base is refined in step 3 from 1 to 0, however, in step 4, the refined based is not used. That means this refinement strategy is always using initial prediction for refinement, what is the advantage and disadvantage of this?

Following R2.Q2 response, why comparison results from response figure differ so much between deepchopper and other methods.

Following R2.Q3 respons for the first part related to validating performance of internal adapter detection in real data, I like the example shown in Figure 5, which quite clearly show that this is a internal adapter, I wonder the sequences before the low base-quality, is a real artefact or fusion reads, and in general, what is the prevalence of the presence of such internal adapter in this real dataset to get an idea of how big the issue is in real data? Related to the second question, as to the justification of using a genomic language model over alignment-based methods, a comparison of the genomic language model vs alignment-based methods (not necessarily minimap2, but some methods build upon minimap2 alignment), for example, the

recently online paper Breakinator tool (Heinz, Jakob M., Matthew Meyerson, and Heng Li. "Detecting Foldback Artifacts in Long Reads." bioRxiv (2025): 2025-07.) and flair-fusion (which aims to identify fusion by removing chimeric reads). It would be worth discussion to highlight the advantage of using genomic language models in this context. Following up to R2 Q4, the response does not explain how different thresholds of this classification decision. In the context of 2 labels, the sum of the 2 adapter class probability would always be 1, that is, the classification of 1 label probability being greater than the other label is still 0.5 threshold. Please clarify and state scenarios where the 2 probabilities does not sum to 1, cause it's hard to understand why in the case of 2 labels, the sum would not be 1. Related to this, response figure 6 is very clean indeed, I wonder is there some kind overfitting or testing data is too simple? Please elaborate for challenging scenarios.

(Remarks on code availability)

Version 2:

Reviewer comments:

Reviewer #3

(Remarks to the Author)

The authors have thoroughly and satisfactorily addressed all of the comments and concerns I raised. I sincerely appreciate their hard work and patience.

(Remarks on code availability)

We would like to thank the reviewers **R1** and **R2** for their thorough review of our manuscript. Please find our point-by-point response below.

R1.General This manuscript introduces DeepChopper, a genomic language model specifically designed to mitigate chimera artifacts in nanopore direct RNA sequencing. The authors leverage HyenaDNA, an efficient genome language model utilizing long-range implicit convolution, to generate output embeddings. This is coupled with a multi-layer perceptron (MLP) and a sliding window-based probability prediction mechanism, enhanced by majority voting for robust predictions. As the first genomic language model tailored for long-read sequencing data, this work marks a significant advancement in transcriptomics. It addresses a critical challenge in the field, showcasing its utility in improving data quality for applications such as transcript annotation and gene fusion detection. The study is innovative, methodologically sound, and a valuable contribution to genomic research. Below, I highlight the manuscript's strengths and suggest revisions to further improve its clarity and impact.

Strength:

1. DeepChopper is the first genomic language model tailored for long-read sequencing data. It provides an end to end pipeline for chimera affected DNA sequence task. Its ability to handle sequences up to 32K nucleotides, combined with its integration of sequencing quality scores, ensures precise detection of adapter sequences at single-nucleotide resolution.
2. The tool is rigorously validated across multiple platforms, including ONT and PacBio, and different chemistries (RNA002 and RNA004). Its robustness and practical impact on reducing false positives in gene fusion detection are well-documented.
3. The method is comparatively faster than traditional methods due to Hyena's long range efficient capabilities and efficient computational complexity.
4. Improvements in transcript annotation and artifact reduction have strong implications for advancing transcriptomics research.

Response We sincerely thank the reviewer for their thoughtful and constructive feedback on our manuscript introducing DeepChopper. We appreciate the recognition of our work's significance and innovation in addressing chimera artifacts in nanopore direct RNA sequencing. Below, we provide point-by-point responses to the reviewer's suggestions.

R1.Q1 Sliding Window Rationale: While the 21-nucleotide sliding window approach achieves an effective balance between smoothing and sensitivity, further explanation of how this parameter was optimized would strengthen the methodology section. Clarifying whether alternative window sizes were tested and how their performance compared would provide additional insights.

Response In response to the reviewer's comment, we conducted additional tests using multiple window sizes to optimize our approach. These results are presented in Response Figure 1 and have been incorporated into the revised manuscript as a **new Extended Data Fig. 3**. We have also added detailed explanations of these tests in the revised manuscript (highlighted sections). As shown in Response Figure 1, we systematically evaluated window sizes of 11, 21, 31, 41, and 51 nucleotides to determine the optimal parameter. In Response Figure 1a, the red bars represent the

Response Figure 1 (Manuscript Extended Data Fig. 4) **Effect of window size on chimeric alignment detection and read fragmentation.** (a) Analysis of different sliding window sizes (11, 21, 31, 41, and 51 nucleotides) showing the percentage of cDNA-supported chimeric alignments (red bars) in VCaP. Higher percentages indicate better support. (b) Distribution of the number of segments per read after trimming (x-axis) for window sizes 11 (gray) and 21 (pink), shown on a logarithmic scale (y-axis). Data represents sampling of 1M reads from VCaP dataset. Window size 21 maintains similar detection sensitivity to window size 11 while producing significantly fewer fragmented reads.

ratio of supporting chimeric alignments by direct cDNA data, with higher percentages indicating better support. Window sizes 11 and 21 showed the highest supporting ratios (47.5% and 47.4%, respectively), with larger windows showing progressively declining support (46.3%, 44.9%, and 43.3% for window sizes 31, 41, and 51). However, Response Figure 1b demonstrates a critical advantage of window size 21 over window size 11. While both detect similar percentages of chimeric alignments, window size 21 produces significantly fewer segments per read after trimming. This indicates that window size 21 maintains more contiguous predicted sequences compared to window size 11, which frequently fragments reads into 4-7 segments. This over-segmentation with window size 11 can potentially introduce artifacts and complicate downstream analysis. Window size 21 therefore represents the optimal balance point in our testing - maintaining the high detection sensitivity of smaller windows while preserving read continuity. Importantly, we recognize that optimal parameters may vary depending on specific dataset characteristics. For this reason, we have implemented a flexible approach by providing the parameter `-smooth-window` in our software, which allows users to adjust the window size according to their specific data requirements. Additionally, we have provided a detailed guide to change the `-smooth-window` and other command-line parameters in response to R1.Q6.

R1.Q2 Other baseline models: Given that other Genome based language models exist (such as DNABERT, DNABERT2, The Nucleotide Transformers), authors might include results from these models as baselines or discussion of these models to better understand the capabilities of Deep-Chopper (in terms of accuracy).

Response We appreciate the reviewer’s suggestion to include comparisons with existing genomic

language models. In response, we have added a comprehensive discussion in the revised manuscript (highlighted) outlining the key differences between DeepChopper and general-purpose genomic language models. As clarified, DeepChopper is specifically designed to address the unique challenges of long-read sequence analysis at single-nucleotide resolution. While models such as DNABERT [1], DNABERT2 [2], and Nucleotide Transformer [3] are valuable for a broad range of genomic tasks, their context length limitations—approximately 512 bp for DNABERT, 10,000 bp for DNABERT2, and 6,000 bp for Nucleotide Transformer—make them suboptimal for long-read applications. In contrast, DeepChopper processes sequences up to 32 kilobases, a context length specifically tailored to accommodate most full-length mRNA transcripts.

Given these fundamental architectural differences, direct performance comparisons are not appropriate. DeepChopper’s design further enables single-nucleotide resolution, a critical feature for base-level precision that general-purpose models are not optimized to deliver. The revised discussion now clearly delineates DeepChopper’s unique strengths in the context of long-read sequence analysis, while acknowledging the broader utility of existing genomic language models.

R1.Q3 Handling of Long Reads: The manuscript notes that sequences exceeding 32K nucleotides are excluded from analysis. Additional details on the prevalence of such sequences in datasets and the implications of this constraint would help users understand the tool’s potential limitations.

Response Thanks for this important comment regarding the handling of ultra-long reads. We have revised the manuscript (highlighted) to provide a comprehensive explanation of the 32K context length limitation and its practical implications.

The 32K limit was introduced based on both architectural and biological considerations. Architecturally, it reflects the context window limitation of the underlying HyenaDNA model [4]. While extending this limit is technically possible, doing so would substantially increase computational demands without clear benefits for the vast majority of use cases.

Biologically, the manuscript now clarifies that human mRNA transcripts have a median length of 1.5-2kb with a peak around 2,065 bp [5], and even complex genes rarely exceed 10–15 kb. Notably, the longest known transcripts, such as titin (TTN), range from 20–25 kb—well within our 32K limit. We also cite nanopore RNA-seq data showing that the maximum aligned read lengths are typically 21 kb, with most reads falling well below this threshold[6].

These updates should provide readers with a clear understanding of the rationale and relevance of the 32K context limit for most long-read transcriptomic applications.

R1.Q4 Performance with RNA004 Chemistry: The reduced efficacy of DeepChopper with RNA004 compared to RNA002 (21% vs. 91% reduction in chimeric alignments) is an important observation. Including a discussion on differences in adapter designs or sequencing chemistry and their potential impact on performance would add depth to the analysis.

Response We appreciate the reviewer highlighting this important difference in performance between RNA002 and RNA004 chemistries. In response, we have expanded our manuscript (highlighted) to address this important observation. Notably, DeepChopper was trained exclusively on RNA002 data, which was the predominant sequencing chemistry available during model development. Despite this training limitation, we deliberately included RNA004 evaluation to assess DeepChopper’s zero-shot transfer capability to newer chemistries without additional fine-tuning. The observed difference in chimeric alignment reduction (91% for RNA002 vs. 21% for RNA004) reflects both the model’s training bias toward RNA002 patterns and fundamental differences in the sequencing chemistries themselves.

RNA004 represents a significant advancement in Oxford Nanopore’s chemistry, specifically designed to produce cleaner reads with fewer artifacts, including chimeric sequences [7]. This inherent improvement in the RNA004 protocol means that raw RNA004 data already contains substantially fewer chimeric artifacts than RNA002 data, giving DeepChopper less opportunity for further improvement. In other words, the smaller relative improvement in RNA004 should be contextualized by the already superior quality of the input data. We have added this explanation to the manuscript (highlighted) to provide readers with a more comprehensive understanding of the performance differences and to highlight DeepChopper’s zero-shot generalization ability across different sequencing chemistries.

R1.Q5 Computational Resource Documentation: Providing a table summarizing runtime, memory, and GPU/CPU requirements for datasets of varying sizes would be highly beneficial for researchers planning large-scale analyses.

Response We thank the reviewer for this valuable suggestion. In response, We have expanded the documentation of computational resource in the revised manuscript by including a **new Extended Data Fig. 4** (Response Figure 2) that illustrates the scaling of runtime and memory requirements across different dataset sizes. As shown in Response Figure 2, we benchmarked DeepChopper’s performance on four dataset sizes (0.1M, 0.5M, 1M, and the full 9M reads) to provide researchers with clear expectations for planning their analyses. The results show that DeepChopper can process a full 9-million-read dataset from the VCaP cell line in approximately **5 hours**.

We have also included a detailed breakdown of resource usage for each processing step:

1. **FASTQ Conversion:** This initial stage shows minimal resource requirements for smaller datasets but scales to 25 minutes and 40 GB CPU memory for the full 9M read dataset.
2. **Adapter Prediction:** This most computationally intensive stage benefits significantly from GPU acceleration, requiring 3 hours and 42 minutes on our full dataset with 33 GB GPU memory and 71 GB CPU memory.
3. **Post-Processing:** This stage scales linearly, requiring 55 minutes and 41 GB CPU memory for the full dataset.

The benchmarks were conducted using high-performance infrastructure (16 CPU cores, dual NVIDIA A100 GPUs), and we have included these specifications in the **Methods** section (highlighted) to provide complete transparency regarding the computing environment. These details have been incorporated into the new “**Computational Benchmarks**” subsection under Methods. These comprehensive benchmarks demonstrate that DeepChopper remains practical for large-scale sequencing projects, efficiently processing millions of reads within hours while maintaining reasonable memory requirements.

Response Figure 2 (Manuscript Extended Data Fig. 3) **Computational performance metrics across different data sizes from the VCaP cell line dRNA-seq.** (a) Runtime analysis showing processing time requirements for different pipeline stages (FASTQ Conversion, Prediction, Post-Processing) and total runtime across four dataset sizes : subsampled (0.1M, 0.5M, 1M) and full (9M) reads derived from the VCaP dRNA-seq. As data size increases, prediction time becomes the dominant component, with the full dataset requiring approximately 5 hours of total processing time. (b) Memory usage comparison between CPU and GPU implementations across the same data sizes. The prediction stage shows consistently higher memory requirements, with CPU memory usage for prediction reaching approximately 70 GB for the larger datasets. All measurements include error bars representing standard deviation from three runs.

R1.Q6 Generalization Guidelines: Including recommendations for parameter tuning in diverse datasets or scenarios would enhance the tool’s applicability across different research contexts.

Response We thank the reviewer for this valuable suggestion. In response, We have created a dedicated “Parameter Optimization Guidelines” (<https://github.com/ylab-hi/DeepChopper/blob/main/documentation/parameters.md>) section in our GitHub documentation to clarify DeepChopper’s applicability across different research contexts. This documentation provides detailed recommendations for optimizing four key command-line parameters used in the post-processing step:

1. Sliding Window Size (*-smooth-window*): We recommend the default value of 21 nucleotides for most applications, as it optimally balances detection sensitivity and read continuity as shown in our comparative analysis (Response Figure 1).
2. Minimum Interval Size (*-min-interval-size*): This post-smoothing parameter (default: 13) can be adjusted based on adapter characteristics, with lower values increasing sensitivity to short adapter fragments.
3. Maximum Process Intervals (*-max-process-intervals*): This parameter (default: 4) limits adapter regions processed per read and can be reduced to decrease fragmentation or increased for complex libraries.

4. **Minimum Read Length (*-min-read-length*):** This final filtering parameter (default: 20) prevents the generation of extremely short fragments and can be adjusted based on the application’s requirements.

These revisions directly address the reviewer’s suggestions and contribute to improving the overall clarity and utility of the manuscript. We thank the reviewer again for their thoughtful feedback, which has helped refine and strengthen the work.

R2.General This manuscript presents DeepChopper, a deep-learning-based tool designed to identify and remove chimera artifacts in Nanopore direct RNA sequencing (dRNA-seq). The authors propose that sequencing artifacts, particularly internal adapters, introduce errors that affect downstream analyses such as transcript annotation and gene fusion detection. DeepChopper leverages HyenaDNA, a genomic language model originally trained on DNA sequences and integrates sequencing quality scores to improve adapter detection. The study evaluates DeepChopper using synthetic training data and real dRNA-seq datasets, claiming a significant reduction in chimeric reads compared to ONT’s Dorado basecaller.

Response We appreciate the reviewer’s thoughtful and constructive feedback on our manuscript presenting DeepChopper. Below, we provide detailed responses to each point raised.

R2.Q1 The manuscript does not clearly explain why a DNA-trained genomic language model (HyenaDNA) is appropriate for RNA sequencing data. RNA has different sequence structures, error profiles, and biological contexts than DNA. Since HyenaDNA was trained specifically on human DNA, it likely requires fine-tuning to adapt to RNA sequencing artifacts. The authors should provide a stronger theoretical justification for why this model is expected to work and discuss whether any modifications were made to improve its performance on RNA data. In addition, if non-human dRNA-seq data were to be analysed, DeepChopper, although not evaluated, will likely to be fail.

Response We thank the reviewer for raising this important point regarding the use of a DNA-trained genomic language model (HyenaDNA [4]) for RNA sequencing data. We provide the following clarifications:

- **Adapter Sequences:** The adapter sequences that we detect are synthetic DNA sequences ligated to RNA molecules during library preparation (as discussed in Ref.3 [8] of the manuscript). Since these adapters are DNA-based, a DNA-trained model is appropriate for this task.
- **Model Transferability:** HyenaDNA is designed to learn nucleotide-level genomic patterns, which are transferable to RNA sequences due to the shared nucleotide composition between DNA and RNA. Genomic language models capture contextual relationships between nucleotides, making them effective for both DNA and RNA analysis (as detailed in Ref.8 [9] of the original manuscript).
- **Fine-Tuning on RNA Data:** To ensure optimal performance, we extensively fine-tuned HyenaDNA using RNA-specific data, allowing it to adapt to the unique error profiles and structural characteristics of Nanopore dRNA-seq. This fine-tuning process enables the model to effectively learn RNA-specific patterns while preserving its core nucleotide understanding (see “**Training Data Preparation**” in the Methods section).

Furthermore, **R2** expressed concern that “if non-human dRNA-seq data were to be analyzed, DeepChopper will likely fail.”

We thank the reviewer for raising this important point. While the reviewer expressed concern that DeepChopper may not generalize to non-human dRNA-seq data, our original manuscript already included an analysis demonstrating its applicability beyond human datasets.

As presented in Extended Data Fig. 2 of the original submission (now updated and relabeled as **Extended Data Fig. 5** in the revised manuscript), we applied DeepChopper to dRNA-seq data from the **F121-9 mouse embryonic stem cell line** provided by the LRGASP consortium. In this analysis, DeepChopper effectively removed chimeric read artifacts that lacked support from cDNA-based long reads across multiple sequencing platforms, confirming its utility in non-human species.

To further strengthen this point and address the reviewer’s concern related to **R2.Q7**, we expanded the figure in the revised manuscript to include new benchmarking results using **Dorado adapter trimming followed by DeepChopper**, as shown in **Extended Data Fig. 5** and Response Figure 3 below. These results demonstrate that DeepChopper consistently removes unsupported chimeric alignments in both Dorado trimmed and untrimmed read sets.

Together, these results support the generalizability of DeepChopper to non-human dRNA-seq data without requiring species-specific retraining.

Response Figure 3 (Manuscript Extended Data Fig. 5) **Chimeric alignments from direct RNA sequencing (dRNA-seq) of the F121-9 cell line (mouse), evaluated for support using additional Oxford Nanopore Technologies (ONT) and Pacific Biosciences (PacBio) sequencing data with different protocols. DeepChopper-involved methods reduce unsupported chimeric alignments across all methods compared to Dorado with adapter trimming. The bar colors indicate chimeric alignments supported by additional sequencing data (red) and those lacking support (grey).**

R2.Q2 DeepChopper is introduced as a novel method, but the manuscript does not compare it to existing adapter trimming and chimera detection tools such as Porechop, Pychopper, or ONT’s Dorado with adapter trimming enabled (rDNA-seq data was base-called with “–no-trim” option). Without direct comparisons in terms of accuracy, precision, recall, F1-score, and runtime, it is unclear whether DeepChopper offers meaningful advantages. A benchmarking analysis is necessary to determine whether DeepChopper outperforms or complements these existing methods.

Response We appreciate the reviewer’s comments regarding the need for benchmarking and comparison of DeepChopper against existing adapter trimming and chimera detection tools. We would like to respectfully clarify that DeepChopper addresses challenges specific to Oxford Nanopore Technologies (ONT) **direct RNA sequencing (dRNA-seq)**—a protocol that is fundamentally distinct from those targeted by the tools mentioned.

1. Protocol-Specific Relevance

DeepChopper is specifically designed to identify and trim adapter sequences in ONT **direct RNA sequencing** data. In contrast, tools such as **PyChopper** [10] are optimized for **cDNA-based protocols** and use models trained on cDNA adapter sequences, which are not present in dRNA-seq datasets. Therefore, a direct comparison is not appropriate, as PyChopper is not built to process or accurately identify dRNA-seq adapters.

2. Limitations of Legacy Tools

Porechop, another tool cited, is no longer actively maintained (last update over eight years ago) and was developed prior to the introduction of ONT’s dRNA-seq technology [11]. Our analysis demonstrates that Porechop is unable to detect adapters in dRNA-seq reads, making a direct comparison with DeepChopper infeasible (see Response Figure 4).

3. Benchmarking on Synthetic dRNA-seq Data

To address the concern, we benchmarked DeepChopper’s performance against a range of adapter trimming tools, including those explicitly mentioned by the reviewer (**PyChopper**, **Porechop**) as well as an additional, more recent extension tool, **Porechop_ABI** [12]. While Porechop_ABI represents an advancement over the original Porechop, none of these tools are specifically designed for the characteristics of ONT **direct RNA sequencing (dRNA-seq)** data.

We performed benchmarking using a **synthetic dRNA-seq dataset** generated in the original manuscript (N = 60,000 reads), which includes known adapter sequences introduced at defined positions. This design allows for robust performance evaluation in terms of precision, recall, and F1-score. As shown in Response Figure 4, **all existing tools**, including **Porechop_ABI**, **failed to accurately detect dRNA-seq adapter sequences**. Their performance metrics were negligible, indicating a fundamental incompatibility with the dRNA-seq protocol.

In contrast, **DeepChopper accurately identified and trimmed both terminal and internal adapters**, highlighting its unique capability to handle the complexities of direct RNA reads. These findings reinforce that current tools are insufficient for adapter detection in dRNA-seq, and underscore the need for a purpose-built solution such as DeepChopper.

4. Additional Benchmarking with Dorado

We have also included a detailed comparison with **Dorado (with adapter trimming enabled)** in our response to **R2.Q7**, where we show that DeepChopper offers superior sensitivity and specificity for detecting dRNA-seq adapters, particularly in complex cases involving chimeric reads.

Response Figure 4 (Manuscript Extended Data Fig. 1) **Performance evaluation in a held-out test ($N = 60,000$) dataset showing Recall, Precision, and F1 values for DeepChopper, Pychopper, Porechop, and Porechop_ABI.**

We have updated the manuscript to include these benchmarking results as **Extended Data Fig. 1** (Response Figure 4) and added **Extended Data Table 1** (Response Table 1), which provides a comparative summary of the capabilities and limitations of current tools versus DeepChopper. These results support the conclusion that existing tools are not well-suited for adapter trimming in dRNA-seq, underscoring the need for a dedicated solution.

Response Table 1 (Manuscript Extended Data Table 1) **Summary of Adapter Trimming Tools for analyzing dRNA-seq data**

Adapter trimming tool	dRNA-seq terminal adapter trimming	dRNA-seq internal adapter trimming	Trimming existing dRNA-seq datasets (post-basecalling)
Porechop [11]	×	×	×
Porechop_ABI [12]	×	×	×
Pychopper [10]	×	×	×
Dorado [13]	✓	×	×
DeepChopper (This study)	✓	✓	✓

Note: ✓ indicates the tool supports this functionality; × indicates the tool does not support this functionality.

We hope these clarifications and new data address the reviewer’s concerns regarding the novelty and benchmarking of DeepChopper.

R2.Q3 The training data were created by manually inserting adapters into reads and identifying adapter regions using minimap2 soft-clipping. This process assumes that all 3’ soft-clipped regions

correspond to adapter sequences, which may not be true. Furthermore, chimeric reads were artificially generated, which does not necessarily reflect real ONT sequencing artifacts. The authors should validate their training data approach by analyzing real dRNA-seq datasets to determine whether internal adapters appear at similar frequencies and distributions. In addition, the authors used minimap2 (a long-read aligner) to generate true training data. Why don't use alignment-based method directly? How the genomic language model will be better than alignment-based methods? Basically, all the benchmarks (e.g. counting chimeric reads, etc.) were all based on alignment results. This needs to be justified.

Response We thank the reviewer for the detailed comments and the opportunity to clarify our training data generation methodology and the rationale behind our modeling approach.

1. Refinement in Training Data Generation Using Biological Context

The reviewer notes that our training data were created by manually inserting adapters and identifying adapter regions using minimap2 soft-clipping, which may not fully capture the complexity of real dRNA-seq artifacts. We appreciate this concern and would like to clarify that **we did not assume all 3' soft-clipped regions to be adapter sequences**. Instead, our method incorporates a biologically informed refinement:

We specifically anchored on polyA tails—a reliable indicator of transcript termination in dRNA-seq—to guide the identification of downstream adapter sequences. Only soft-clipped regions of low base-quality that followed a polyA stretch were considered candidate adapter segments. This strategy substantially reduces the risk of falsely labeling endogenous soft-clipped sequences as adapter-derived, ensuring higher-quality training labels.

To support this strategy, we provide an example from real dRNA-seq data (Response Figure 5). In this example, a read corresponding to the GAS5 gene does not terminate at the expected polyA region (highlighted in red) but continues with a sequence segment of notably lower base quality (labeled “Low base quality region”). These low-quality regions extend into the soft-clipped part of the alignment, strongly indicating the presence of adapter-containing chimera artifacts. The figure clearly demonstrates how these artifacts display a consistent sequence pattern—well-aligned regions followed by polyA stretches, which are then followed by poorly aligned, low-quality segments (shown in the gray shaded area in Response Figure 5) that correspond to the adapter sequences and subsequent incorrectly captured fragments. This pattern recognition approach significantly reduces the risk of misclassifying non-adapter regions as adapters and ensures our training data more accurately reflects the biological context of direct RNA sequencing.

Response Figure 5 Visualization of adapter-containing chimeric reads in dRNA-seq of the VCaP cell line. This alignment visualization shows characteristic features of chimeric artifacts in dRNA-seq data. Individual reads (horizontal lines) display a consistent pattern: well-aligned sequences (colored bases) followed by internal polyA stretches (labeled “polyA”), which are immediately followed by regions of significantly lower base quality (labeled “Low base quality region”). These low-quality regions, extending into the soft-clipped portions (gray shaded area), represent adapter sequences and incorrectly captured fragments. This distinctive pattern—internal polyA followed by low-quality regions—provides a reliable signature for discriminating technical artifacts from genuine biological chimeras.

2. Justification for Using a Genomic Language Model Over Alignment-Based Methods

The reviewer also questions why alignment-based methods (e.g., minimap2) were not used directly for detecting chimeric artifacts, especially since alignment results were employed in benchmarking.

We clarify that while **alignment is useful for initial labeling and benchmarking, it is not sufficient for accurate artifact detection.** This is because:

- (a) **Not all chimeric alignments are artifacts.** Some represent biologically meaningful events, such as trans-splicing or gene fusion events.
- (b) **Alignment-based tools lack the context to distinguish artifact from biology.** Without modeling sequence composition, adapter motifs, or base quality patterns, traditional aligners cannot differentiate true gene fusions from chimera artifacts.
- (c) Our **genomic language model captures the latent patterns and statistical properties** of genuine transcripts versus synthetic chimeric sequences, including nucleotide distribution, quality scores, and adapter motifs.
- (d) This modeling approach enables **DeepChopper to selectively filter artificial chimeras while preserving biologically relevant reads**, which is crucial in accurate transcript annotation and minimizing false-positive gene fusion calls (see Fig. 3 in the manuscript).

Thus, while alignment was used in training and validation, our approach offers a learned, sequence-aware alternative that is capable of generalizing beyond rule-based alignment heuristics.

We have revised the manuscript to provide further details on these points and clarify the polyA-anchored strategy in the **Methods: Training Data Preparation** section.

R2.Q4 DeepChopper classifies bases as adapters if the probability score is greater than 0.5, but the manuscript does not justify this threshold. If probability scores are clustered around 0.5, the classification may be unreliable. The authors should provide a distribution of probability scores and assess whether different cutoffs affect performance. An ROC curve or precision-recall analysis would help determine the optimal threshold for adapter detection.

Response We thank the reviewer for pointing out the ambiguity regarding the classification threshold used by DeepChopper. We agree that the description in the original manuscript—specifically the reference to a fixed threshold of 0.5—may have led to confusion.

To clarify, **DeepChopper does not use a fixed threshold of 0.5 for classification**. Instead, it employs a *softmax*-based multi-class classification approach, where the model outputs per-base probabilities for two classes: adapter and non-adapter. During inference, classification is performed by selecting the class with the higher probability at each nucleotide position, i.e., a base is assigned to the adapter class if $P(y_i = \text{adapter}) > P(y_i = \text{non-adapter})$, and vice versa. This approach inherently avoids reliance on a fixed threshold and ensures consistent interpretation of probabilistic outputs, especially in the context of mutually exclusive class labels.

We have updated the manuscript to correct this discrepancy and provide a more accurate description of the classification head. Specifically, we have revised the relevant sentence in the *Language Model Architecture* subsection of the **Methods** to reflect the use of softmax rather than sigmoid, and to clarify the decision rule applied during inference.

In response to the reviewer’s suggestion, we have also performed additional analyses to characterize the model’s prediction confidence and assess the reliability of its classifications. We now include a new figure (Response Figure 6) that visualizes the distribution of predicted probabilities for the adapter and non-adapter classes on a held-out test set ($N = 60,000$ reads). These analyses show that:

1. For true adapter bases (Response Figure 6a), the model assigns high adapter probabilities (mostly > 0.8), and correspondingly low non-adapter probabilities.
2. For true non-adapter bases (Response Figure 6b), the model confidently predicts high non-adapter probabilities (> 0.8), with low adapter probabilities (< 0.2).
3. The overall probability distributions are strongly bimodal, indicating decisive predictions and minimal ambiguity between classes.

This sharp separation in probability space demonstrates robust feature learning and generalization.

We sincerely appreciate the reviewer’s valuable feedback, which has helped us improve both the clarity and completeness of our manuscript.

Response Figure 6 (Manuscript Extended Data Fig. 2) **Prediction probability distributions of DeepChopper for the held-out test dataset ($N = 60,000$).** (a) Distribution of prediction probabilities for sequences with ground truth adapter classification. Red bars represent the probability of adapter prediction, while gray bars show the probability of non-adapter prediction. The count (y-axis) is shown in millions of sequences (10^6 scale). (b) Distribution of prediction probabilities for sequences with ground truth non-adapter classification. Red bars indicate the probability of adapter prediction, while gray bars show the probability of non-adapter prediction. The count (y-axis) is shown in tens of millions of sequences (10^7 scale). Both distributions demonstrate strong polarization toward correct classification probabilities, indicating the model’s high confidence in distinguishing between adapter and non-adapter sequences.

R2.Q5 The manuscript introduces a Quality Block to incorporate sequencing base quality scores but it does not clearly explain how this improves model performance. Similarly, the sliding window refinement approach lacks detailed methodology, including how window size and majority voting criteria were selected. The authors should provide an ablation study testing the impact of the Quality Block and a sensitivity analysis for the sliding window parameters.

Response We appreciate the reviewer’s thoughtful comments and the opportunity to clarify the methodology and impact of both the Quality Block and the sliding window refinement procedure. We have revised the manuscript to include additional details and supporting analyses, which are now highlighted in the updated version.

Quality Block Contribution The Quality Block is designed to incorporate base-level sequencing quality scores into DeepChopper’s prediction pipeline, enhancing its robustness in noisy regions of the input sequence—particularly near ambiguous adapter boundaries, which exhibit low overall quality (See Response Figure 5).

In response to the reviewer’s suggestion, we performed an ablation study using an independent dataset ($N = 100,000$ reads). As shown in Response Table 2, incorporating the Quality Block improved the F1 score from 0.97 to **0.99**, indicating a notable improvement in classification performance attributable to this component. This enhancement reflects the ability of quality signals to disambiguate difficult predictions, particularly in low-confidence regions. We have added the training setup to the revised Methods section and included the results of this ablation study as **Extended Data Table 2** in the revised manuscript.

Response Table 2 (Manuscript Extended Data Table 2) Ablation Study Results for Quality Block

Model Configuration	F1 Score
With Quality Block	0.99
Without Quality Block	0.97

Sliding Window Refinement To further improve prediction smoothness and reduce isolated misclassifications, DeepChopper applies a post-hoc sliding window refinement step based on majority voting. We have now expanded the Methods section to detail both the rationale and parameter selection strategy.

We evaluated multiple window sizes (ranging from 11 to 51 nucleotides) on VCaP, balancing detection sensitivity with read continuity (See **R1.Q1**). As shown in Response Figure 1a, window sizes of 11 and 21 performed similarly in terms of raw prediction support (47.5% and 47.4%, respectively). However, as illustrated in Response Figure 1b, a window size of **21** significantly reduced the number of read fragments introduced by trimming, thereby preserving better overall sequence integrity while maintaining sensitivity. Based on this analysis, window size 21 was selected as the default.

To ensure flexibility for different datasets and sequencing characteristics, we have implemented the *-smooth-window* parameter in the command-line interface, allowing users to customize the window size as needed. Detailed documentation for this feature is available in DeepChopper’s online user guide and in our response to Reviewer 1, Comment 6 (See **R1.Q6**).

Majority Voting Mechanism The sliding window refinement applies a straightforward majority voting strategy over a centered window of size W , defined as follows:

$$y_i = \begin{cases} 1 & \text{if } \sum_{j=i-k}^{i+k} p_j > \frac{W}{2} \\ 0 & \text{otherwise} \end{cases}$$

Here, y_i is the final refined prediction at nucleotide position i , $k = \frac{W-1}{2}$ is the window radius, and p_j represents the initial binary prediction at position j , where 1 indicates adapter and 0 indicates non-adapter. This approach smooths the basewise predictions by consolidating local prediction context, reducing isolated false positives and negatives.

We thank the reviewer for prompting these improvements, which have enhanced the clarity and rigor of our methodology. The relevant sections in the manuscript have been updated and highlighted accordingly.

R2.Q6 The manuscript states that DeepChopper has 4.6 million parameters, suggesting it is computationally efficient but it does not provide any runtime or memory benchmarks. There is no information on how long DeepChopper takes to process a dataset, what GPU/CPU resources are required, or how it scales with large sequencing datasets. A computational efficiency evaluation is necessary to determine whether DeepChopper is practical for large-scale sequencing projects.

Response We thank the reviewer for this important suggestion. To address the concern regarding computational efficiency, we have conducted a comprehensive benchmarking analysis and expanded the relevant documentation in the revised manuscript.

As shown in Response Figure 2, we evaluated DeepChopper’s runtime and memory usage across datasets ranging from 0.1 million to 9 million reads. For the full 9 million-read VCaP dataset, DeepChopper completes processing in approximately **5 hours**. We also provide a detailed breakdown of resource requirements for each stage of the pipeline:

1. **FASTQ Conversion:** 25 minutes, peak **40 GB** CPU memory
2. **Adapter Prediction:** 3 hours and 42 minutes, peak **33 GB** GPU memory and **71 GB** CPU memory
3. **Post-processing and Sliding Window Refinement:** 55 minutes, peak **41 GB** CPU memory

These benchmarks were performed on a system with **16 CPU cores** and **two NVIDIA A100 GPUs**. The hardware specifications and software environment have been added to the Methods section (highlighted) for full transparency.

This analysis demonstrates that DeepChopper scales efficiently to large datasets, with runtime and memory usage remaining practical for large-scale sequencing applications. The full benchmarking results, including scalability trends, are also discussed in our response to Reviewer 1, Comment 5 (See R1.Q5).

R2.Q7 The validation dRNA-seq data were processed using Dorado with the “–no-trim” option, which preserves adapters and likely increases the observed chimeric read count. This choice could artificially inflate the problem DeepChopper aims to solve, making the model appear more effective than it actually is. The authors should repeat their analysis using Dorado with adapter trimming enabled to assess whether DeepChopper still provides a significant advantage.

Response We thank the reviewer for raising this important concern. The potential for inflated chimeric read counts due to the use of Dorado with the `--no-trim` option is well taken, and we appreciate the opportunity to clarify and further substantiate DeepChopper’s effectiveness.

In response to the reviewer’s suggestion, we extended our benchmarking and downstream analyses by incorporating results from Dorado with adapter trimming enabled, both alone and in combination with DeepChopper. In our analysis of VCaP dRNA-seq data, we observed that Dorado alone—regardless of whether trimming was applied—was insufficient to eliminate spurious chimeric alignments. In contrast, DeepChopper substantially reduced unsupported chimeric events, whether applied independently or following Dorado trimming, as validated by orthogonal cDNA sequencing (Response Figure 7).

Response Figure 7 (Manuscript Fig. 2d) **Chimeric alignments (in thousands) for VCaP dRNA-seq reads processed by Dorado with and without adapter trimming, Dorado with adapter trimming followed by DeepChopper, and DeepChopper. DeepChopper-involved methods including Dorado with adapter trimming followed by DeepChopper and DeepChopper greatly reduce chimeric alignments not supported by direct cDNA sequencing.**

To further generalize these findings, we performed an expanded analysis across six additional human cell lines (A549, HCT116, HepG2, K562, MCF7, and WTC11) as well as the F121-9 mouse embryonic stem cell line. As shown in Response Figure 8, Response Figure 9, and Response Figure 3, DeepChopper consistently reduced chimeric artifacts across all cases, including when applied after Dorado’s adapter trimming. These results reinforce that DeepChopper offers distinct and additive benefits beyond what is achieved through adapter trimming alone.

Response Figure 8 (Manuscript Fig. 2g) The number of chimeric alignments (in thousands) for A549, HCT116, HepG2, K562, and MCF7 cell lines processed by Dorado with adapter trimming, Dorado with adapter trimming followed by DeepChopper and DeepChopper. DeepChopper-involved methods including Dorado with adapter trimming followed by DeepChopper and DeepChopper consistently reduce chimeric alignments not supported by cDNA sequencing across all cell lines.

Response Figure 9 (Manuscript Fig. 2h) Chimeric alignments from dRNA-seq of the WTC11 cell line, evaluated for support using additional ONT and PacBio sequencing data with different protocols. DeepChopper-involved methods including Dorado with adapter trimming followed by DeepChopper and DeepChopper reduce unsupported chimeric alignments across all methods compared to Dorado with adapter trimming.

We also evaluated the impact of DeepChopper on isoform-level transcript detection using VCaP cells. As illustrated in Response Figure 10, applying DeepChopper following Dorado trimming led to a greater number of detected transcripts across multiple isoform classification categories compared to Dorado trimming alone. This supports the complementary role of DeepChopper in enhancing downstream transcriptomic interpretation, even when applied after standard trimming procedures.

Response Figure 10 (Manuscript Fig. 3b) The bar plot shows the number of transcripts (in thousands) across different isoform classification categories. DeepChopper-processed reads result in a higher number of transcripts compared to Dorado-trimmed reads. The inset details the isoform classification scheme.

Furthermore, we repeated our gene fusion analysis using Dorado-trimmed reads followed by DeepChopper. As shown in Response Figure 11, fusion detection was markedly improved in both DeepChopper-based pipelines—with or without prior trimming—demonstrating greater concordance with gene fusions identified from matched short-read RNA-seq data. This underscores DeepChopper’s ability to eliminate spurious fusion calls while preserving true, biologically meaningful fusion events.

Response Figure 11 (Manuscript Fig. 3c) Detected gene fusions from Dorado adapter-trimmed reads and DeepChopper-processed reads using Dorado with adapter trimming followed by DeepChopper and DeepChopper. Gene fusions identified from short-read RNA-seq were used to validate fusion events detected from dRNA-seq.

Collectively, these findings confirm that DeepChopper’s effectiveness is not contingent upon the inclusion of untrimmed adapter sequences. Its capacity to reduce chimeric artifacts, enhance isoform detection, and improve gene fusion calling remains robust across diverse datasets and preprocessing workflows. We have incorporated these updated analyses and figure references into the revised manuscript to directly address this concern.

References

- [1] Yanrong Ji et al. “DNABERT: Pre-Trained Bidirectional Encoder Representations from Transformers Model for DNA-language in Genome”. In: *Bioinformatics* 37.15 (Aug. 9, 2021). Ed. by Janet Kelso, pp. 2112–2120. ISSN: 1367-4803, 1367-4811. DOI: [10.1093/bioinformatics/btab083](https://doi.org/10.1093/bioinformatics/btab083).
- [2] Zhihan Zhou et al. *DNABERT-2: Efficient Foundation Model and Benchmark for Multi-Species Genome*. 2023. DOI: [10.48550/arXiv.2306.15006](https://doi.org/10.48550/arXiv.2306.15006). arXiv: [2306.15006](https://arxiv.org/abs/2306.15006) [cs, q-bio]. Pre-published.
- [3] Hugo Dalla-Torre et al. “Nucleotide Transformer: building and evaluating robust foundation models for human genomics”. In: *Nature Methods* (2024), pp. 1–11.
- [4] Eric Nguyen et al. “Hyenadna: Long-range genomic sequence modeling at single nucleotide resolution”. In: *Advances in neural information processing systems* 36 (2024).
- [5] Inês Lopes et al. “Gene size matters: an analysis of gene length in the human genome”. In: *Frontiers in Genetics* 12 (2021), p. 559998.
- [6] Rachael E Workman et al. “Nanopore native RNA sequencing of a human poly (A) transcriptome”. In: *Nature methods* 16.12 (2019), pp. 1297–1305.
- [7] Charlotte Hewel et al. “Direct RNA sequencing enables improved transcriptome assessment and tracking of RNA modifications for medical applications”. In: *bioRxiv* (2024), pp. 2024–07.
- [8] Martin A Smith et al. “Molecular barcoding of native RNAs using nanopore sequencing and deep learning”. In: *Genome research* 30.9 (2020), pp. 1345–1353.
- [9] Gonzalo Benegas et al. *Genomic Language Models: Opportunities and Challenges*. 2024. arXiv: [2407.11435](https://arxiv.org/abs/2407.11435) [q-bio.GN]. URL: <https://arxiv.org/abs/2407.11435>.
- [10] *epi2me-labs/pychopper: cDNA read preprocessing*. Github. URL: <https://github.com/epi2me-labs/pychopper>.
- [11] Ryan R. Wick et al. “Completing bacterial genome assemblies with multiplex MinION sequencing”. In: *Microbial Genomics* 3.10 (Oct. 2017). ISSN: 2057-5858. DOI: [10.1099/mgen.0.000132](https://doi.org/10.1099/mgen.0.000132). URL: <http://dx.doi.org/10.1099/mgen.0.000132>.
- [12] Quentin Bonenfant, Laurent Noé, and H el ene Touzet. “Porechop_ABI: discovering unknown adapters in Oxford Nanopore Technology sequencing reads for downstream trimming”. In: *Bioinformatics Advances* 3.1 (2023), vbac085.
- [13] Oxford Nanopore PLC. *Dorado*. <https://github.com/nanoporetech/dorado>. 2023.

We would like to thank the reviewers **R1** and **R3** for their thorough review of our . Please find our point-by-point response below.

R1.General The authors have addressed all the comments and concerns I raised in the previous review. I recommend that the paper be published in its current form.

Response We sincerely thank **R1** for continued support and recommendation for publication. We appreciate the recognition of our revisions and the thoroughness of the review process.

R3.General The describes genomic language model based method for detecting chimeric reads in direct RNA sequencing. Overall, the is well written and thoughtfully organised, where the comments are also well addressed. I have a few questions remain after reading the , which either arise from reading the or following up to previous question which I felt more can be done to address those questions.

Response We thank **R3** for their thoughtful and constructive feedback on our revised . We appreciate the recognition that the is well-written and well-organized, and that we have addressed previous comments comprehensively. Below, we provide detailed responses to the remaining questions, which we hope will fully address the reviewer’s concerns.

R3.Q1 The focuses on direct RNA seq, I wonder how generalizable is the approach to cDNA seq data, would a lot effort be needed? It would be good if authors could discuss for this application given there are a lot cases where Nanopore cDNA samples are generated instead.

Response We thank the reviewer for raising this important point regarding generalizability to cDNA-seq. We agree that cDNA-seq is widely used, and we have revised the Discussion to clarify platform-specific considerations and future extensions.

1. DeepChopper’s Design and dRNA-seq Chimeras

DeepChopper, in its current implementation, is tailored to dRNA-seq because the dominant chimera formation mechanism in dRNA-seq produces **internal adapter sequences**, which serve as a molecular signature of non-biological chimeras. Across six human cell lines, mouse cell, and two ONT chemistries (Manuscript Figures 2–3, Extended Data Figures 8-11, **R3.Q9**), we consistently observe that dRNA-seq chimeras contain these internal adapters. This enables two key capabilities: (i) reliable artifact identification, since internal adapter presence cannot arise from biological processes; and (ii) accurate sub-read recovery by splitting at adapter boundaries, which preserves bona fide transcriptomic rearrangements [1] and increases usable read yield.

2. Mechanistic Difference: cDNA-seq and Template Switching

Nanopore cDNA-seq produces a fundamentally different class of chimeras. These arise mainly from **reverse-transcriptase template switching**, in which chimeric junctions are formed by direct template-to-template joining and therefore lack internal adapter sequences [2–6]. Prior studies, including Schulz et al. (2021), have shown that such RT-mediated artifacts can closely mimic biological events—for example, putative exon splicing events detected only in cDNA-seq but absent in dRNA-seq—underscoring that these artifacts have distinct molecular signatures that DeepChopper’s adapter-based strategy is not designed to detect.

3. Potential for Adaptation:

While DeepChopper as a detection-and-recovery tool is specific to dRNA-seq, the Genomic Language Model (GLM) that forms the foundation of DeepChopper is general and not tied to adapter features. This GLM-based framework can, in principle, be trained to recognize RT-mediated cDNA-seq artifacts by learning their sequence-context patterns (e.g., microhomology, local sequence motifs associated with RT template switching). We are actively exploring these directions. However, distinguishing RT artifacts from genuine biological events in cDNA-seq remains challenging because, unlike the definitive internal adapter signal left by artifacts in dRNA-seq, RT template switching creates chimeric molecules through direct template-to-template joining, thus leaving no molecular tag (e.g., an internal adapter) for easy identification. Current cDNA-focused methods therefore rely primarily on alignment-based rules [7], and we compare these approaches with DeepChopper’s adapter-based strategy in Response **R3.Q10**.

We appreciate the reviewer’s suggestion and have expanded the Discussion section to clarify (i) why DeepChopper currently targets dRNA-seq, (ii) how the foundational GLM is extensible to cDNA-seq artifact modeling, and (iii) what challenges and methodological considerations arise for future applications to cDNA-seq.

R3.Q2 The novelty of this approach, is the application of genomic language models, I agree. However, the advantage of this model vs conventional non-large language model is not discussed enough for why this approach should be favored. One point related is the testing data being too easy, which is shown as clear bimodal distribution as presented in response figure 6, any challenging scenarios that could benefit from deepchopper but not other methods?

Response We thank the reviewer for this insightful question regarding the advantages of GLM and the perceived simplicity of the test data. We have revised the manuscript to clarify why GLM offer capabilities beyond conventional approaches and to highlight challenging scenarios where DeepChopper provides clear benefits.

1. Why Genomic Language Models Offer Substantive Advantages

DeepChopper’s core novelty is not only applying an GLM, but enabling a new class of artifact detection that cannot be addressed by rule-based or alignment-based methods (**R3.Q8** and **R3.Q10**). These advantages arise from three technical properties:

(a) Error-tolerant pattern learning for unknown or corrupted adapter sequences In dRNA-seq, internal adapters are heavily corrupted because (i) the chemical adapters for dRNA-seq are not publicly documented, and (ii) DNA adapters are mis-called by RNA-trained basecallers. Conventional methods rely on exact matching, k-mer counting, or alignment heuristics and therefore fail when adapters are error-rich, truncated, or shifted.

In contrast, the GLM learns probabilistic sequence patterns even under high error conditions. This ability is what enables DeepChopper to identify > 90% of artifacts in some datasets—even when the true adapter sequence is never provided. We show that per-base quality integration further boosts F1 to 0.99 (Manuscript Extended Data Table 3), demonstrating the model’s ability to leverage multi-modal contextual signals.

(b) Alignment-free detection eliminates blind spots of alignment-dependent methods

Alignment-based artifact detection inherently fails in three challenging but common scenarios:

1. Artifacts that map as single alignments (no chimeric signature; invisible to fusion/structural filters) (Response Figure 1)
2. Reads that fail to map due to adapter contamination, thus never reaching downstream filters (Response Figure 1)
3. Artifacts in repetitive or structurally complex regions, where aligners generate ambiguous mappings

DeepChopper operates directly on raw sequences, circumventing these limitations entirely. This allows recovery of sub-reads from contamination-heavy sequences that all alignment-dependent filters would simply discard (R3.Q10).

(c) Preserving genuine biological rearrangements that conventional methods mistakenly remove

Because conventional methods evaluate alignment patterns (breakpoints, distances, orientations), they cannot distinguish artifacts from bona fide biological chimeras, which often produce identical mapping signatures (R3.Q10). This leads to valuable biological reads being removed.

Empirically, this distinction is decisive:

- Breakinator decreases cDNA-supported rearrangements (4.8%, Response R3.Q10)
- DeepChopper increases support nearly ten-fold (5% → 47 – 49%) (Manuscript Figure 2)
- Gene-fusion false positives fall by 89% while novel transcripts nearly double (Manuscript Figure 3)

These gains reflect a unique conceptual advantage: **DeepChopper removes artifacts because they contain adapter-like sequence patterns, not because they merely “look chimeric” after alignment.**

2. Addressing the Reviewer’s Concern: Is the Test Data Too Easy?

The bimodal distribution shown in Response Figure 6 reflects that some artifacts contain long adapter fragments, making them easily separable. However, our analyses also include multiple challenging scenarios where conventional tools fail but DeepChopper succeeds:

(a) Short, fragmented, or partially mis-called adapters Many internal adapters are only 10–18 bp long and heavily corrupted (Manuscript Figure 2a, 2e, and 2f). Rule-based filters cannot detect these; DeepChopper reliably does (R3.Q8 and R3.Q10).

(b) Non-chimeric alignments with internal adapters Up to 33–45% of adapter-containing reads map as single alignments (Response Figure 1). No alignment-based tool can flag these, yet they are effectively removed by DeepChopper (R3.Q10).

Response Figure 1 (Manuscript Extended Data Figure 4) **Percent of internal adapter-containing reads that are non-chimeric** Percentage of internal adapter-containing reads that do not produce chimeric alignments across five human cell lines (RNA002) processed by Dorado with trim followed by DeepChopper. Between 33–45% of adapter-containing reads map as single alignments or fail to map, making them invisible to chimeric alignment-based artifact detection.

(c) Complex transcripts with true rearrangements In genes with extensive alternative splicing or rearranged exons, conventional methods over-filter and remove biologically valid events. DeepChopper preserves these while removing artifacts, providing a net increase in usable biological signal. (R3.Q10)

(d) High-error-rate or low-quality chemistry datasets We included two sequencing chemistries with different error profiles. DeepChopper maintains performance in high-error regimes where exact-matching methods fail entirely. (R3.Q8)

These scenarios illustrate that the GLM’s advantages are most pronounced in the hardest regions of the distribution, not the easy ones.

3. Summary of Why This Approach Should Be Favored

DeepChopper’s GLM foundation enables three capabilities that no conventional method can replicate:

1. Detect error-corrupted internal adapters without prior knowledge of sequences
2. Operate completely alignment-free, eliminating blind spots where alignment-based filters fail
3. Distinguish artifacts from true biological rearrangements, increasing biological signal instead of diminishing it

These points are now emphasized in the revised Discussion to clarify why DeepChopper enable a fundamentally different and more powerful solution to adapter-bridged chimera removal.

R3.Q3 (more specific) Line 059-064: The chimera artifacts in dRNA-seq resulted from ligation (biases? I assume) or software missing open pore signal. The citation seems to describe more in a multiplexing context. I wonder is there any paper discussing this for regular ONT dRNA-seq instead? Also, the part software missing open pore signals is from paper in 2020, specifically referring to MinKNOW, which has been outdated, have the authors checked whether this is still a problem with the recent dRNA-seq?

Response We thank the reviewer for these important observations about citation currency and technology advances. The reviewer has identified a significant literature gap that our work addresses.

1. Are adapter-bridged chimeras documented for regular (non-multiplexed) ONT dRNA-seq?

The reviewer is correct that prior publications describing ligation- or pore-related chimeras primarily focus on multiplexing settings or earlier chemistries. To our knowledge, **no peer-reviewed study has characterized internal-adapter chimeras in standard, non-multiplexed dRNA-seq libraries.** This gap motivated our work.

Our analyses—performed on non-multiplexed samples, using current Dorado basecalling and both RNA002 and RNA004 chemistries—show that internal-adapter artifacts are systematic and reproducible (Response Table 2):

- RNA002: 0.67–1.62% of total reads contain internal adapters across six human cell lines
- RNA002: 63–92% of chimeric reads contain internal adapters across six human cell lines
- RNA004: prevalence decreases but persists (0.33% of total reads and 24% of chimeric reads in VCaP), still affecting tens of thousands of reads per run

Thus, even though the literature has not previously reported this phenomenon in non-multiplexed dRNA-seq, our study provides quantitative evidence that these chimeras remain an intrinsic feature of current ONT dRNA workflows.

2. Are older citations still relevant, and do “open pore signal” artifacts persist?

The reviewer is correct that the cited MinKNOW-related discussion originated from 2020 and predates Dorado. We agree this required clarification, and we have revised the manuscript to avoid implying that MinKNOW-specific behavior is responsible for the artifacts we observe today.

Importantly, our results indicate:

- The internal-adapter chimeras we detect persist with Dorado (v0.5.2) and modern RNA004 chemistry.
- These artifacts are not restricted to older Guppy/MinKNOW pipelines.

Our study does not rely on the presence of “open pore signal” errors in current software. Instead, we empirically identify internal-adapter junctions directly in the read sequences, independent of the mechanism by which they arise.

Because no current ONT documentation provides a definitive molecular explanation for these chimeras, we frame the mechanistic statement more cautiously. We now state that:

- internal-adaptor chimeras are empirically observed,
- occur independently of multiplexing,
- and persist even with the latest chemistry and basecalling.

We have revised the manuscript Introduction and Discussion sections to clarify the literature gap, emphasize validation with current technology (Dorado, RNA004), and detail the comprehensive validation framework.

R3.Q4 Related to the handling of long-reads, as questioned by Reviewer 1, 32kb misses a few genes, for example SMAD2, LTRT, TTN (TTN actually is longer than 32kb?), etc? How are these genes dealt with then? And what impact will happen to these genes? In terms of expression levels? The read length referred to citation in the response is rather outdated, and the annotation with respect is also quite old.

Response We thank the reviewer for raising this important question regarding ultra-long transcripts and the 32 kb context window. We have expanded the manuscript to clarify transcriptome-wide coverage using current annotations and to quantify the practical impact on real dRNA-seq data.

1. Empirical Finding: reads > 32 kb are extremely rare in modern dRNA-seq

Across seven datasets spanning RNA002 and RNA004 chemistries, reads exceeding 32 kb represent only **0-0.0032% of all reads** (Response Table 1). Even VCaP RNA004—the largest dataset with improved chemistry enabling longer reads—had only 379 reads out of 11.72 million (0.0032%) exceeding 32 kb. Mean read lengths range 683-1,148 bp; 99th percentile reaches only 2.6-4.7 kb across all datasets.

Response Table 1 (Manuscript Extended Data Table 2) Read Length Statistics

Sample	Reads (M)	Mean (bp)	P95 (bp)	P99 (bp)	Max (bp)	Reads \geq 32kb	%
A549	1.70	907	2,440	3,829	16,246	0	0
MCF7	3.04	715	1,863	3,052	28,802	0	0
HCT116	4.70	889	2,431	3,793	21,656	0	0
K562	3.06	683	1,736	2,619	58,395	2	0
HepG2	1.80	1,148	3,025	4,665	46,077	1	0
VCaP RNA002	9.18	994	2,826	4,399	77,474	1	0
VCaP RNA004	11.72	995	2,784	4,474	225,798	379	0.0032

For the remaining 99.9968% of reads within 32 kb, DeepChopper provides substantial benefits: 62-91% artifact reduction, nearly two-fold transcript discovery, and 89% fewer false gene fusions (Manuscript Figures 2-3). Reads longer than 32 kb are passed through unchanged and remain available for downstream analyses.

2. Updated transcriptome analysis using current annotation (Ensembl GRCh38.115, July 2025)

To address the reviewer’s concern about outdated transcript length references, we reanalyzed the full set of protein-coding transcripts in Ensembl GRCh38.115 (released July 2025):

- Median: \sim 2.7 kb
- 95th percentile: \sim 8.7 kb

- 99th percentile: ~ 14 kb
- Transcripts >32 kb: $<0.03\%$ of all protein-coding transcripts

Response Figure 2 (Manuscript Extended Data Figure 1) shows this distribution with 32 kb threshold marked. Over 99.97% of transcripts fall below this limit.

Response Figure 2 (Manuscript Extended Data Figure 1) **Distribution of transcript length for protein-coding genes.** Analysis of all protein-coding transcripts from Ensembl GRCh38.115 (released July 2025) shows that $>99.97\%$ of transcripts are below the 32 kb threshold (marked with vertical dashed line). The distribution is highly skewed toward shorter transcripts, with median length of ~ 2.7 kb.

3. Handling of ultra-long genes (e.g., TTN, SMAD2)

For the very small number of transcripts exceeding 32 kb:

- DeepChopper does not modify these reads to avoid truncation of biologically meaningful regions
- Expression quantification remains unaffected, as these reads pass through intact and are counted normally downstream.
- Biological impact is negligible, given that such reads constitute $< 0.003\%$ of real data and $< 0.03\%$ of annotated transcripts.
- Artifact considerations remain minimal, as ultra-long reads rarely carry adapter contamination and do not appreciably influence transcript-level summaries.

4. Future Directions

While the current 32 kb window is computationally optimal for typical dRNA-seq workflows, the underlying HyenaDNA architecture in DeepChopper can, in principle, scale to ultra-long contexts

approaching 1M tokens [8]. For specialized applications where complete modeling of exceptionally long transcripts is critical—such as full-length analysis of genes like TTN—we plan to develop and release extended-context versions of DeepChopper. These future models will support substantially larger context windows while retaining the algorithmic efficiencies required for transcriptome-scale processing.

We have revised the manuscript Results and Discussion sections to acknowledge this limitation using current annotations (Ensembl GRCh38.115, July 2025) and added Extended Data Figure 1 and Table 2.

R3.Q5 Following Review 1 Q4 regarding RNA004, for practicality, it would be good if the authors could show fine-tuned results of deep-chopper for RNA004 in terms of performance to be more relevant in direct RNA sequencing context.

Response We thank the reviewer for this practical suggestion. We demonstrate strong RNA004 performance with both original and fine-tuned models.

1. Performance with Current RNA004 Chemistry

Response Figure 3 (Manuscript Extended Data Figure 10) compares six conditions using cDNA-seq validation. Key findings:

Strong zero-shot cross-chemistry generalization: The original RNA002-trained model achieved 21-27% artifact reduction on RNA004 data while preserving all $\sim 7,000$ cDNA-supported biological rearrangements. This demonstrates robust generalization despite different chemistry-specific error profiles—adapter-bridged chimera formation mechanisms remain consistent across chemistries.

Fine-tuning provides marginal improvement: RNA004-optimized fine-tuning achieved 24-32% artifact reduction—an additional 3-4% improvement over zero-shot performance. Both models preserve identical cDNA-supported alignments ($\sim 7,000$), confirming selective removal of adapter-bridged artifacts while maintaining genuine biological events.

Response Figure 3 (Manuscript Extended Data Figure 10) **RNA004 performance: original vs fine-tuned DeepChopper.** Chimeric alignments in VCaP RNA004 across six conditions. Red: cDNA-supported biological events; Grey: unsupported artifacts. Baseline (Dorado only): ~29-30K total alignments. Original DeepChopper: 21-27% artifact reduction. Fine-tuned DeepChopper: 24-32% artifact reduction. Both models preserve all cDNA-supported biological rearrangements.

Complementary with Dorado trimming: Performance remains nearly identical regardless of Dorado trim order, confirming complementary functions: Dorado removes 3' end adapters during basecalling; DeepChopper detects internal adapters in chimeric reads (R3.Q8 and R3.Q11).

2. Practical Recommendations

Original model (recommended for most users): Strong zero-shot performance (21-27% artifact reduction) with no additional setup. Robust across both RNA002 and RNA004 chemistries.

Fine-tuned RNA004 model (specialized applications): Marginal benefit (3-4% additional improvement). Recommended for RNA004-exclusive large-scale studies requiring maximum artifact reduction.

Both models available at <https://github.com/ylab-hi/DeepChopper> with selection guidance in documentation. Fine-tuning used 300,000 VCaP RNA004 reads (70/20/10 train/validation/test split). Same architecture and hyperparameters as RNA002 training, adapting to RNA004-specific error profiles. Processing time: ~9 hours on two NVIDIA A100 GPUs. We have added this analysis to Results section and Extended Data Figure 10, providing optimized models for both RNA002 and RNA004 platforms.

R3.Q6 Following Reviewer 1 Q5 response, the authors demonstrated it in different data size, in terms of practicality, an data of size 20/25 million would also be interested to be included for this guidance.

Response We thank the reviewer for this practical suggestion. We benchmarked DeepChopper on 23 million reads—a merged dataset combining five cell lines (A549, HCT116, HepG2, K562, MCF7) from our validation framework (**R3.Q3**).

1. Performance at 23M Scale

Response Figure 4 (Manuscript Extended Data Figure 6) shows benchmarks across five data sizes (0.1M, 0.5M, 1M, 9M, 23M reads) with triplicate measurements.

Processing time (23M reads, two NVIDIA A100 GPUs):

- Total: 10.6 hours
- FASTQ conversion: 23 min
- Prediction: 8.5 hours
- Post-processing: 1.7 hours

Near-linear scaling with dataset size confirms efficient computational performance without algorithmic bottlenecks. Prediction dominates at scale (80%), expected for sequential neural network processing of long reads.

Memory usage (stable across all sizes):

- CPU: 70-93 GB (prediction stage peak)
- GPU: 34-56 GB (stage-dependent)

Stable memory footprint indicates no fundamental barriers to processing beyond 23M reads—memory determined by batch size, not dataset size.

Response Figure 4 (Manuscript Extended Data Figure 6) **Computational performance metrics across different data sizes.** (a) Runtime analysis showing processing time requirements for different pipeline stages (FASTQ Conversion, Prediction, Post-Processing) and total runtime across five data sizes: subsampled VCaP datasets (0.1M, 0.5M, 1M reads), full VCaP direct RNA sequencing (dRNA-seq) dataset (9M reads), and merged large-scale dataset (23M reads combining A549, HCT116, HepG2, K562, and MCF7). Runtime scales near-linearly with data size. As data size increases, prediction time becomes the dominant component, requiring approximately 5 hours for the 9M dataset and 10.6 hours for the 23M dataset. (b) Memory usage comparison between CPU and GPU implementations across the same data sizes. The prediction stage shows consistently higher memory requirements. CPU memory usage ranges from 70-93 GB and GPU memory from 34-56 GB across larger datasets, with stable memory footprint indicating no fundamental barriers to processing substantially larger datasets. All measurements include error bars representing standard deviation from three technical replicates.

The biological benefits—62-91% artifact reduction enabling accurate transcript annotation and gene fusion detection (**R3.Q2**)—justify this computational investment for high-quality transcriptome profiling.

We have added this 23M benchmark to the Results section and updated Extended Data Figure 6, providing concrete performance expectations for large-scale studies.

R3.Q7 Figure 1 c). Sliding window worth more explanation for better understanding. For example, from step 3 to step 4, the bottom 3rd base is refined in step 3 from 1 to 0, however, in step 4, the refined based is not used. That means this refinement strategy is always using initial prediction for refinement, what is the advantage and disadvantage of this?

Response We thank the reviewer for carefully examining the refinement strategy in Figure 1c. The reviewer is correct that the refinement procedure always uses the *initial* predictions rather than the refined values from previous steps. This behavior is intentional and reflects both biological and computational design requirements.

1. Rationale: Biological Constraints Require Non-Iterative Refinement

DeepChopper’s core task is not simply to classify reads as chimera artifact or non-artifact, but to **precisely localize internal adapter boundaries** so that sub-reads can be recovered with

single-nucleotide accuracy. Adapter boundary precision is essential because:

- A 1–2 nt boundary shift disrupts reading frames
- Mislocalized boundaries truncate or distort biological exons
- Downstream transcript reconstruction becomes invalid

To satisfy these constraints, we designed a refinement strategy that evaluates each position *independently* using a sliding window of the **original model predictions**, not sequentially refined values. This avoids the risk that early refinement errors accumulate and propagate across the read.

2. How the Sliding Window Operates

For each base at position i , the refinement step uses:

$$p_{i-k}, \dots, p_i, \dots, p_{i+k}$$

where p_j are the **initial** model probabilities (default window size = 21 bp).

As the window slides from step 3 to step 4:

- The window shifts by one nucleotide
- The refined value from step 3 is *not* carried forward
- Position $i + 1$ is re-evaluated solely from the initial predictions

Thus, each refined prediction is computed from a fixed reference distribution, not influenced by previous refinement decisions.

3. Advantages of This Non-Iterative Refinement

Prevents error cascades. If one position is mis-refined, that error does not contaminate subsequent positions. This isolation is essential for accurate boundary detection.

Preserves single-nucleotide accuracy. Independent evaluation ensures that each base reflects local evidence only—critical for adapter boundary calls where even single-nucleotide shifts compromise recovered sub-reads.

Efficient and parallelizable. Because positions are independent, the refinement layer can be computed in parallel across all bases, enabling DeepChopper to scale to tens of millions of reads (see R3.Q6).

4. Trade-offs and Mitigation

Trade-off: Independent refinement may leave more variability in noisy regions compared with iterative smoothing approaches.

Mitigation strategies used in DeepChopper:

- **Quality-aware predictions** (Quality Block; R3.Q2) increase robustness and raise F1 from 0.97 to 0.99.
- **Length-filtering** removes implausibly short fragments (default minimum = 20 bp; R3.Q11).
- **User-adjustable window size** allows fine-tuning for different datasets or error profiles.

5. Summary

Using only the initial predictions during refinement is a deliberate design choice that ensures (i) error isolation, (ii) single-nucleotide precision critical for adapter boundary detection, and (iii) scalable, parallel computation. We have clarified this strategy in the revised Methods section.

R3.Q8 Following R2.Q2 response, why comparison results from response figure differ so much between deepchopper and other methods.

Response We thank the reviewer for this important question. The dramatic performance differences reflect fundamental capability gaps—existing tools were designed for different problems and lack internal adapter detection functionality.

1. The Unsolved Problem: Internal Adapter Detection

Our validation confirms that internal adapter detection in dRNA-seq was an unsolved computational problem prior to DeepChopper. On held-out test data (60,000 reads: 90% containing adapters split equally between 3' end and internal positions, 10% negative examples without adapters, evaluated at single-nucleotide resolution), existing tools achieved negligible adapter detection performance (Manuscript Extended Data Table 1, Extended Data Figure 2) while DeepChopper achieved $F1 > 0.99$. This reflects capability gaps, not tool quality—**each tool excels at its intended application, but none were designed for internal adapter detection in dRNA-seq.**

Real-world validation across six human cell lines, mouse cells, and two chemistries (**R3.Q3**) confirms this capability gap: adapter-bridged chimeras persist when processed with existing tools, requiring DeepChopper's specialized approach for detection and recovery.

2. Why Existing Tools Cannot Detect Internal Adapters

Dorado (ONT basecaller): Excels at 3' end adapter trimming during basecalling but cannot detect internal adapters. Basecalling proceeds sequentially 5'→3'; 3' end detection identifies end-of-molecule (open pore signal), but internal adapters appear as continuous sequence without distinguishing signals. Our validation (**R3.Q3**, **R3.Q5**, and **R3.Q11**) confirms adapter-bridged chimeras persist in RNA002 and RNA004 when processed with Dorado—3' end trimming does not address internal adapters.

Pychopper: Designed exclusively for cDNA-seq strand-switching primers (SSP, VNP) absent in dRNA-seq [9]. Detects only 3' end primers for orientation—no internal detection capability. Performs excellently for intended cDNA applications.

Porechop: Developed 2017, abandoned 2018 [10]. Attempts internal detection via exact sequence matching against predefined database. Fails because: (1) database lacks dRNA adapter sequences; (2) exact matching cannot tolerate sequencing errors; (3) DNA adapters mis-called by RNA-trained basecallers corrupt exact matches.

Porechop_ABI: Uses k-mer frequency analysis to infer adapters [11]. Fails because: (1) sequencing errors corrupt k-mer patterns; (2) internal adapters appear at low frequency—insufficient signal for k-mer detection.

3. Why DeepChopper Succeeds: Genomic Language Model Advantages

As detailed in R3.Q2, DeepChopper’s success is rooted in the unique capabilities of the Genomic Language Model (GLM) architecture, which specifically addresses the failure points listed above:

- **Error-Tolerant Pattern Learning:** The pre-trained GLM learns **probabilistic sequence characteristics** that distinguish biological RNA from adapter artifacts. This capability tolerates sequencing errors and basecaller miscalls that cripple exact-matching methods.
- **Alignment-Free Operation:** DeepChopper works directly on the raw base-called sequence, eliminating systematic biases. This is critical because not all internal artifacts produce chimeric alignments (some map as single alignments or fail to map entirely) and are thus invisible to alignment-based detection methods.
- **Long-Range Context:** The DeepChopper’s 32K token window allows it to scan complete transcripts (up to 32 kb) to detect adapters at any variable internal position, unlike local heuristics.
- **Preservation of Genuine Biology:** By using the adapter’s definitive presence as the marker, DeepChopper can selectively remove only artifacts while preserving genuine biological rearrangements (e.g., trans-splicing), a distinction rule-based alignment methods cannot make. This distinction led to DeepChopper increasing cDNA support from 5% to 47–49% across datasets.

In summary, the performance disparity reflects the fact that DeepChopper is the first tool designed with the advanced computational capabilities necessary to solve the unique technical challenges of internal adapter detection in dRNA-seq. We have expanded the Discussion section to clarify these points.

R3.Q9 Following R2.Q3 responds for the first part related to validating performance of internal adapter detection in real data, I like the example shown in Figure 5, which quite clearly show that this is a internal adapter, I wonder the sequences before the low base-quality, is a real artefact or fusion reads, and in general, what is the prevalence of the presence of such internal adapter in this real dataset to get an idea of how big the issue is in real data?

Response Thank you for this insightful question regarding (1) the prevalence of internal adapters in real datasets and (2) whether the sequences flanking the low–base-quality region in Figure 5 represent true biological events or artifacts. We provide a consolidated explanation below.

1. Internal Adapters Are a Systematic and High-Impact Artifact in Real dRNA-seq Data

DeepChopper detected internal adapters in 0.67-1.62% of all reads across RNA002 datasets and 0.33% in RNA004 (Response Table 2), representing 15,690-148,452 affected reads per experiment. Importantly, when restricting analysis to reads that align as chimeras, internal adapters account for the majority of chimeric reads:

1. 63–92% of all chimeric reads in RNA002 datasets
2. 24% in RNA004 (reduced but still substantial)

Thus, in a typical 10-million–read experiment, tens to hundreds of thousands of reads contain internal adapters even after current best-practice Dorado trimming. This shows that internal adapters remain a systematic, not sporadic, artifact in modern ONT dRNA-seq workflows.

Response Table 2 (Manuscript Extended Data Table 4) Internal Adapter Prevalence Across Datasets

Sample	All Reads			Chimeric Reads Only		
	With Internal Adapters (A)	Total (B)	% (A/B)	With Internal Adapters (C)	Total (D)	% (C/D)
A549	15,690	1,703,697	0.92	10,553	12,803	82.43
MCF7	20,340	3,039,468	0.67	11,115	17,646	63.00
HCT116	57,122	4,697,299	1.22	37,823	46,800	80.81
K562	29,436	3,061,722	0.96	19,289	23,214	83.09
HepG2	22,530	1,797,922	1.25	14,331	16,921	84.69
VCaP RNA002	148,452	9,177,422	1.62	98,878	107,265	92.18
VCaP RNA004	38,878	11,714,520	0.33	6,891	29,144	23.65

Note: Total reads from Dorado (v0.5.2) with trim. Internal adapters detected by DeepChopper after Dorado processing.
Note: VCaP RNA002 and RNA004 demonstrate that adapter-bridged chimeras persist across chemistries (discussed in R3.Q3 and R3.Q5), though RNA004 shows reduced prevalence.

2. Are the Sequences Flanking the Low-Quality Region True Biological Events?

We re-examined the flanking sequences associated with the low-quality region shown in Figure 5 (Previous response; now Manuscript Figure 3e). As illustrated in Response Figure 5 (now Manuscript Extended Data Figure 13), read 3b2292e9-43e5-4e40-87d9-ccc23897377c (the read displayed in Figure 5) contains two high-confidence flanking regions. BLAT alignment confirms that the upstream (5') segment (positions 1–154) maps to chr14 within the RPS29 gene with **98% identity**, and the downstream (3') segment (positions 230–722) maps to chr11 within the COX8A gene with **96% identity** (Response Figure 5). These results demonstrate that both flanking sequences correspond to genuine biological RNA derived from distinct loci.

In contrast, the intervening segment (positions 155–229) does **not** align to the reference genome and exhibits hallmarks of an internal adapter artifact (Response Figure 5 highlighted in red). Across the dataset, regions identified as adapters by DeepChopper consistently show:

- **Low base quality:** Median $Q \sim 10$ throughout the adapter region (Response Figure 5; Manuscript Figure 2e)
- **Poor genome alignment:** Median BLAT identity ~ 0.3 (Response Figure 5; Manuscript Figure 2f)
- **Poly-A context:** Poly-A sequences immediately upstream of the adapter (Response Figure 5; highlighted in green)

Response Figure 5 (Manuscript Extended Data Fig. 13) **Representative internal adapter detection with base quality visualization.** Representative read from VCaP RNA002 processed with Dorado trim followed by DeepChopper, demonstrating internal adapter detection (highlighted in red, positions 155-229) and sub-reads recovery (blue box and purple box). (upper panel) BLAT alignment of upstream sequence (blue box, positions 1–154) to chr14 (RPS29 gene) with 0.98 identity, confirming genuine biological RNA. (middle panel) Full read visualization showing internal adapter and sub-reads. Base quality scores (Q scores) shown as color-coded bars: yellow indicates high quality ($Q > 40$), dark blue indicates low quality ($Q < 10$). The adapter region shows characteristic poly-A sequences upstream (highlighted in green) and lower base quality compared to flanking biological sequences. The adapter region (75 bp) shows no matches found to the reference genome. Sequences before and after the adapter represent genuine biological RNA from different transcripts artificially joined during library preparation or basecalling. Read ID: 3b2292e9-43e5-4e40-87d9-ccc23897377c. (bottom panel) BLAT alignment of downstream sequence (purple box, positions 230–722) to chr11 (COX8A gene) with 0.96 identity, confirming genuine biological RNA from a different transcript.

Together, these features confirm the internal sequence detected by DeepChopper is an adapter, while both flanks represent true biological RNA. This read therefore reflects an adapter-bridged chimera, where transcripts from RPS29 (chr14) and COX8A (chr11) were artificially joined during library preparation or basecalling.

Without DeepChopper, such reads are often discarded entirely—losing valid biological signal from both transcripts—or incorrectly interpreted as a bona fide RPS29–COX8A gene fusion. By detecting internal adapters and splitting reads at precise boundaries, DeepChopper recovers both transcript fragments as independent sub-reads rather than removing the molecule altogether.

As established in **R3.Q8**, no existing tool detects internal adapters—Dorado’s 3’ end trimming does not reduce internal adapter-bridged artifacts (Manuscript Figures 2-3, Extended Data Figures 8-11). These systematic errors remain undetected in current dRNA-seq workflows, propagating through transcript annotation, gene fusion detection, and expression quantification. We have added Extended Data Table 4 and Extended Data Figure 13 to the Results and Discussion section.

R3.Q10 Related to the second question, as to the justification of using a genomic language model over alignment-based methods, a comparison of the genomic language model vs alignment-based methods (not necessarily minimap2, but some methods build upon minimap2 alignment), for example, the recently online paper Breakinator tool (Heinz, Jakob M., Matthew Meyerson, and Heng Li. “Detecting Foldback Artifacts in Long Reads.” bioRxiv (2025): 2025-07.) and flair-fusion (which aims to identify fusion by removing chimeric reads). It would worth discussion to highlight the advantage of using genomic language models in this context.

Response We thank the reviewer for this insightful question, which encourages a direct comparison between genomic language models and alignment-based artifact detection methods.

1. Direct Comparison With Alignment-Based Tools

To address this question, we benchmarked DeepChopper against Breakinator [7] using the VCaP RNA002 dataset, with cDNA sequencing serving as ground truth validation. The result (Response Figure 6; now Manuscript Extended Data Figure 3) shows the following:

- DeepChopper removes 95.1% of unsupported chimeric alignments while increasing the fraction of cDNA-supported biological chimeras from 5.8% (baseline) to 48.7%.
- Breakinator reduces total chimeric alignments by 61.6%, but the number of cDNA-supported biological chimeras drops to 4.8%, which is lower than the baseline. This indicates indiscriminate removal of both artifacts and genuine biological events.
- Breakinator detects zero RT-mediated foldbacks, as expected, because direct RNA sequencing does not use reverse transcription.

These results clearly demonstrate that DeepChopper selectively removes adapter-bridged artifacts while preserving biologically meaningful rearrangements.

Response Figure 6 (Extended Data Figure 3) **Comparison of chimeric alignment reduction strategies in VCaP RNA002 dRNA-seq data.** Stacked bar plot showing chimeric alignments (in thousands) for three processing pipelines: Dorado with adapter trimming (baseline), Dorado with adapter trimming followed by DeepChopper, and Dorado with adapter trimming followed by Breakinator. Gray bars represent unsupported chimeric alignments (likely artifacts); pink bars represent cDNA-supported chimeric alignments (biological events). DeepChopper achieves substantial reduction in unsupported chimeric alignments (95.1%) while preserving cDNA-supported biological chimeras (support rate: 48.7%). In contrast, Breakinator shows moderate reduction in total chimeric alignments (61.6%) but decreases the cDNA support rate to 4.8%, lower than the baseline (5.8%), indicating indiscriminate removal of both artifacts and biological events.

2. Why Alignment-Based Alignment Methods Cannot Distinguish Artifacts from Biology

Fundamental paradigm difference: DeepChopper uses **adapter-based detection**—internal adapter presence definitively indicates library preparation/basecalling artifacts, not biological events. This enables unambiguous classification and selective removal.

Alignment-based methods use **rule-based filtering**—hand-crafted thresholds on alignment patterns (distance > 1 Mb, interchromosomal, opposite strand within 200 bp) [7]. These patterns occur in *both* adapter-bridged artifacts *and* genuine biological rearrangements, making discrimination impossible.

Breakinator limitations for dRNA-seq: Breakinator [7] detects two types of artifacts based on alignment patterns and distance thresholds: (1) RT foldback inversions (opposite strand breakpoints within 200 bp), targeting RT template-switching artifacts absent in dRNA-seq, precisely the RT chimeras discussed in **R3.Q1** that lack internal adapters; and (2) distance-based chimera filtering (intrachromosomal breakpoints > 1Mb, interchromosomal events) that cannot distinguish

adapter-bridged artifacts from genuine biological RNA rearrangements. Neither detection strategy addresses internal adapters, and Breakinator removes entire reads without sub-read recovery, discarding genuine biological sequences flanking adapters.

FLAIR-fusion limitations: Task-specific gene fusion caller [12], not general QC tool. Filtering rules leverage gene annotation and splice sites—information unavailable for other analyses like isoform quantification. No adapter detection capability. We encountered execution failures; issues reported on GitHub (<https://github.com/cafelton/FLAIR-fusion/issues/6>) remain unresolved.

3. Advantages of DeepChopper: Genomic Language Models Over Alignment-Based Methods

As established in **R3.Q2** and **R3.Q8**, DeepChopper’s genomic language model framework delivers critical capabilities beyond the reach of traditional approaches. **Detection via internal adapter signatures provides unambiguous artifact identification:** adapter presence within reads conclusively marks library preparation or basecalling failures rather than biological processes, allowing DeepChopper to retain authentic biological rearrangements that distance-based rules eliminate indiscriminately. **Reference-independent processing** avoids mapping biases and alignment ambiguities (**R3.Q9**, Response Table 2). Key genomic language model features supporting this strategy include: (1) **foundation model training on genomic data** establishes intrinsic patterns of authentic biological sequences[8, 13, 14], supporting discrimination between genuine transcripts and adapter contaminants through learned representations without requiring known adapter motifs; (2) **robust pattern recognition** achieving reliable detection despite degraded sequences and uncharacterized adapter sequences; (3) **base-quality integration** incorporating per-nucleotide confidence scores (Extended Data Table 3); (4) **extended receptive fields** through HyenaDNA’s 32K nucleotide window [8] enabling adapter identification at any internal location; and (5) **nucleotide-level boundary accuracy** via independent positional assessment (**R3.Q7**), critical for successful sub-read extraction.

4. Summary

These tools address fundamentally different problems. DeepChopper performs pre-alignment quality control with internal adapter detection and sub-read recovery in dRNA-seq, while Breakinator targets post-alignment artifact detection based on distance rules in cDNA-seq and FLAIR-fusion provides specialized fusion calling with artifact-aware filtering. For dRNA-seq, DeepChopper fills a critical gap that alignment-based methods cannot address: detecting internal adapters and recovering genuine biological sequences, as established in **R3.Q1**, **R3.Q2**, and **R3.Q8**. We have added this comparison to the Discussion section and Extended Data Figure 3, clarifying DeepChopper’s unique position in dRNA-seq workflows and the complementary nature of different artifact detection approaches.

R3.Q11 Following up to R2 Q4, the response does not explain how different thresholds of this classification decision. In the context of 2 labels, the sum of the 2 adapter class probability would always be 1, that is, the classification of 1 label probability being greater than the other label is still 0.5 threshold. Please clarify and state scenarios where the 2 probabilities does not sum to 1, cause it’s hard to understand why in the case of 2 labels, the sum would not be 1. Related to this, response figure 6 is very clean indeed, I wonder is there some kind overfitting or testing data is too simple? Please elaborate for challenging scenarios.

Response We thank the reviewer for this precise and insightful question. We clarify the probability formulation and expand our discussion of challenging scenarios to address concerns regarding overfitting.

1. Clarifying the Probability Threshold

The reviewer is absolutely correct: for a binary softmax classifier, the two output probabilities always sum to 1. Thus:

$$P(\text{adapter}) + P(\text{non-adapter}) = 1$$

Our decision rule is therefore equivalent to applying a 0.5 threshold on $P(\text{adapter})$. We indeed use a standard softmax formulation—primarily for calibrated confidence scores and extensibility to multi-class settings—but the implicit 0.5 threshold should have been stated explicitly. We have clarified this in the revision.

2. Why the Distribution in Response Figure 6 Looks “Clean”

The clean bimodal distribution is not due to test set simplicity or overfitting. Rather, it reflects a task where the sequence signal at adapter boundaries is strong: internal adapter regions have pronounced base-quality drops and characteristic non-biological sequence patterns that the model reliably learns.

We emphasize that the model’s generalization is supported by extensive real-data validation across multiple chemistries (RNA002 and RNA004) and multiple cell types (A549, MCF7, HCT116, K562, HepG2, and VCaP), as described in our response to **R3.Q2** and **R3.Q2**.

The task is **not** trivial. Below we highlight concrete challenging scenarios observed in real datasets that demonstrate where the model encounters difficulty.

3. Challenging Real-World Scenarios and Model Limitations

Scenario 1: Incomplete detection of 3’ end adapters in multi-adapter reads In reads containing both an internal adapter and a 3’ end adapter, DeepChopper consistently detects the internal adapter but may partially miss the 3’ end adapter. The flanking biological sequence remains intact and is not misclassified. This reflects the model’s prioritization of internal signals, which is appropriate for its intended purpose.

Response Figure 7 (Extended Data Figure 14) shows read c16c6ade-135b-4073-a1d6-5a9c6900bfb2 (VCaP RNA002, Dorado without trim): internal adapter positions 1026-1105 correctly detected, 3’ end adapter partially missed. This occurs when multiple adapters present at different positions.

Response Figure 7 (Extended Data Figure 14) **Challenge Scenario 1: Incomplete 3' terminal adapter detection in multi-adapter read.** Representative read from VCaP RNA002 processed with Dorado without trim followed by DeepChopper (with refinement applied). The read contains both an internal adapter (positions 1026-1105, highlighted in red, correctly detected) and a 3' end adapter that DeepChopper failed to completely detect. Base quality scores (Q scores) shown as color-coded bars: yellow indicates high quality ($Q > 40$), dark blue indicates low quality ($Q < 10$). The internal adapter shows characteristic low quality compared to flanking biological sequences. This scenario demonstrates that when multiple adapters are present, DeepChopper reliably detects internal adapters (its primary function) but may incompletely detect terminal adapters. Read ID: c16c6ade-135b-4073-a1d6-5a9c6900bfb2.

Solution: Combined Dorado-DeepChopper workflow (Response Figure 8) (Extended Data Figure 16). Dorado removes 3' end adapters; DeepChopper detects internal adapters. Complementary functions confirmed in **R3.Q5** and **R3.Q8**.

Response Figure 8 (Extended Data Figure 16) Solution for Challenge Scenario 1: Combined Dorado-DeepChopper workflow. The same representative read (ID: c16c6ade-135b-4073-a1d6-5a9c6900bfb2) processed with Dorado with trim followed by DeepChopper. Dorado successfully removed the 3' end adapter, while DeepChopper detected the internal adapter (highlighted in red, positions adjusted after Dorado trimming). Base quality scores (Q scores) shown as color-coded bars: yellow indicates high quality ($Q > 40$), dark blue indicates low quality ($Q < 10$). The internal adapter shows characteristic low quality. This demonstrates that combining Dorado (3' end adapters) with DeepChopper (internal adapters) addresses complementary problems and resolves the incomplete detection issue shown in Challenge Scenario 1.

Scenario 2: Partial internal detection In reads containing an internal adapter (3' end adapter is trimmed by Dorado), DeepChopper detects part of the internal adapter. Response Figure 9 (Extended Data Figure 15) shows read 0955d980-9c79-48c2-a474-08c8c39cb00f (VCaP RNA002, Dorado with trim): partial adapter detection in highly degraded sequences. Portions of adapter may be missed when error-rich.

Response Figure 9 (Extended Data Figure 15) Challenge Scenario 2: Partial internal adapter detection. Representative read from VCaP RNA002 processed with Dorado trim option followed by DeepChopper (with refinement applied). The read shows partial adapter detection (highlighted in red), demonstrating challenges with degraded or error-rich adapter sequences where DeepChopper may detect only portions of the complete adapter sequence. Base quality scores (Q scores) shown as color-coded bars: yellow indicates high quality ($Q > 40$), dark blue indicates low quality ($Q < 10$). Such partial detections can lead to spurious short fragments, which are addressed through post-processing length filtering. Read ID: 0955d980-9c79-48c2-a474-08c8c39cb00f.

Solution: Post-processing length filtering (default 20 bp minimum, configurable via `--min-read-length`). Removes spurious short fragments while preserving biological sequences. Combined with sliding window refinement (**R3.Q7**), provides quality control ensuring biologically meaningful sub-reads retained.

4. Summary

These challenging scenarios have minimal impact on overall performance:

- **Primary objective achieved:** Internal adapter detection—DeepChopper’s core purpose (**R3.Q1**)
- **Complementary tools:** Dorado (3’ end) + DeepChopper (internal) address both artifact types effectively
- **Biological discovery enabled:** 62-91% artifact reduction, nearly two-fold transcript discovery, 89% fewer false fusions, preserved biological rearrangements (Figures 2-3)
- **Robust generalization:** Consistent performance across cell lines, species, chemistries

Critically, as emphasized in **R3.Q1** and **R3.Q2**, DeepChopper solves a previously unsolved

problem—no existing tool detects internal adapters in dRNA-seq. Strong performance across diverse validation confirms genuine capability. We have added Extended Data Figures 14-16 (challenging scenarios and solutions) to the Discussion section, demonstrating transparency about limitations alongside overall strong performance.

References

- [1] Cong Ma, Mingfu Shao, and Carl Kingsford. “SQUID: Transcriptomic Structural Variation Detection from RNA-seq”. In: *Genome Biology* 19 (Apr. 12, 2018), p. 52. ISSN: 1474-7596. DOI: [10.1186/s13059-018-1421-5](https://doi.org/10.1186/s13059-018-1421-5). PMID: 29650026.
- [2] Zsolt Balázs et al. “Template-Switching Artifacts Resemble Alternative Polyadenylation”. In: *BMC Genomics* 20.1 (Nov. 8, 2019), p. 824. ISSN: 1471-2164. DOI: [10.1186/s12864-019-6199-7](https://doi.org/10.1186/s12864-019-6199-7).
- [3] Laura Schulz et al. “Direct Long-Read RNA Sequencing Identifies a Subset of Questionable Exons Likely Arising from Reverse Transcription Artifacts”. In: *Genome Biology* 22.1 (June 28, 2021), p. 190. ISSN: 1474-760X. DOI: [10.1186/s13059-021-02411-1](https://doi.org/10.1186/s13059-021-02411-1).
- [4] Daniel R Garalde et al. “Highly parallel direct RNA sequencing on an array of nanopores”. In: *Nature methods* 15.3 (2018), pp. 201–206.
- [5] Miten Jain et al. “Advances in nanopore direct RNA sequencing”. In: *Nature methods* 19.10 (2022), pp. 1160–1164.
- [6] Yongji Zou et al. “A Comparative Evaluation of Computational Models for RNA Modification Detection Using Nanopore Sequencing with RNA004 Chemistry”. In: *Briefings in Bioinformatics* 26.4 (July 1, 2025), bbaf404. ISSN: 1477-4054. DOI: [10.1093/bib/bbaf404](https://doi.org/10.1093/bib/bbaf404).
- [7] Jakob M. Heinz, Matthew Meyerson, and Heng Li. *Detecting Foldback Artifacts in Long-Reads*. Sept. 23, 2025. DOI: [10.1101/2025.07.15.664946](https://doi.org/10.1101/2025.07.15.664946). Pre-published.
- [8] Eric Nguyen et al. “Hyenadna: Long-range genomic sequence modeling at single nucleotide resolution”. In: *Advances in neural information processing systems* 36 (2024).
- [9] Camille Sessegolo et al. “Transcriptome Profiling of Mouse Samples Using Nanopore Sequencing of cDNA and RNA Molecules”. In: *Scientific Reports* 9.1 (Oct. 17, 2019), p. 14908. ISSN: 2045-2322. DOI: [10.1038/s41598-019-51470-9](https://doi.org/10.1038/s41598-019-51470-9).
- [10] Ryan R. Wick et al. “Completing bacterial genome assemblies with multiplex MinION sequencing”. In: *Microbial Genomics* 3.10 (Oct. 2017). ISSN: 2057-5858. DOI: [10.1099/mgen.0.000132](https://doi.org/10.1099/mgen.0.000132). URL: <http://dx.doi.org/10.1099/mgen.0.000132>.
- [11] Quentin Bonenfant, Laurent Noé, and H el ene Touzet. “Porechop-ABI: discovering unknown adapters in Oxford Nanopore Technology sequencing reads for downstream trimming”. In: *Bioinformatics Advances* 3.1 (2023), vbac085.
- [12] Colette Felton et al. *Detection of Alternative Isoforms of Gene Fusions from Long-Read RNA-seq with FLAIR-fusion*. July 18, 2023. DOI: [10.1101/2022.08.01.502364](https://doi.org/10.1101/2022.08.01.502364). Pre-published.
- [13] Hugo Dalla-Torre et al. “Nucleotide Transformer: building and evaluating robust foundation models for human genomics”. In: *Nature Methods* (2024), pp. 1–11.
- [14] Zhihan Zhou et al. *DNABERT-2: Efficient Foundation Model and Benchmark for Multi-Species Genome*. 2023. DOI: [10.48550/arXiv.2306.15006](https://doi.org/10.48550/arXiv.2306.15006). arXiv: 2306.15006 [cs, q-bio]. Pre-published.